

# Secondary organic aerosol formation from α-pinene, alkanes and oil sands related precursors in a new oxidation flow reactor

Kun Li, John Liggio, Patrick Lee, Chong Han, Qifan Liu, and Shao-Meng Li

Air Quality Research Division, Environment and Climate Change Canada, Toronto, Ontario M3H 5T4, Canada.

**Correspondence:** John Liggio (john.liggio@canada.ca)

**Abstract.** Oil sands (OS) operations in Alberta, Canada are a large source of secondary organic aerosol (SOA). However, the SOA formation process from OS-related precursors remains poorly understood. In this work, a newly developed oxidation flow reactor (OFR), the Environment and Climate Change Canada OFR (ECCC-OFR), was characterized and used to study the yields and composition of SOA formed from OH oxidation of α-pinene, selected alkanes, and the vapors evolved

from five OS-related samples (OS ore, naphtha, tailings pond water, bitumen, and dilbit). The derived SOA yields from α-pinene and selected alkanes using the ECCC-OFR were in good agreement with those of traditional smog chamber experiments, but significantly higher than those of other OFR studies under similar conditions. The results also suggest that gas-phase reactions leading to fragmentation (i.e., C-C bond cleavage) have a relatively small impact on the SOA yields in the ECCC-OFR at high photochemical ages, in contrast to other previously reported OFR results. Translating the impact of

fragmentation reactions in the ECCC-OFR to ambient atmospheric conditions reduces its impact on SOA formation even further. These results highlight the importance of careful evaluation of OFR data, particularly when using such data to provide empirical factors for the fragmentation process in models. Application of the ECCC-OFR to OS-related precursor mixtures, demonstrated that the SOA yields from OS ore and bitumen vapors (maximum of ~0.6-0.7) are significantly higher than those from the vapors from solvent use (naphtha), effluent from OS processing (tailing pond water) and from the

solvent diluted bitumen (dilbit) (maximum of ~0.2-0.3), likely due to the volatility of each precursor mixture. A comparison of the yields and elemental ratios (H/C and O/C) of the SOA from the OS-related precursors to those of linear and cyclic alkane precursors of similar carbon numbers suggests that cyclic alkanes play an important role in the SOA formation in the OS. The analysis further indicates that the majority of the SOA formed downwind of OS facilities is derived from open-pit mining operations (i.e., OS ore evaporative emissions), rather than from higher volatility precursors from solvent use during

processing and/or tailing management. The current results have implications for improving the regional modeling of SOA from OS sources, for the potential mitigation of OS precursor emissions responsible for observed SOA downwind of OS operations, and for the understanding of petrochemical and alkane derived SOA in general.



## 1 Introduction

Over the last several decades, oil production from unconventional sources has increased significantly and is expected to continue to increase into the future due to its abundant reserves, particularly in North America (Alboudwarej et al., 2006;Mohr and Evans, 2010;Owen et al., 2010). The Alberta oil sands (OS) is a large unconventional crude oil deposit,

which is extracted through both open-pit mining and in-situ steam assisted techniques. Considering the scale of OS oil production, a number of environmental concerns associated with OS air emissions have arisen, including the potential for ecosystem toxicity (Kirk et al., 2014;Harner et al., 2018) and acid deposition (Jung et al., 2011;Makar et al., 2018). Recent field measurements have shown that OS mining and processing facilities are a large source of volatile organic compounds (VOCs) (Simpson et al., 2010;Li et al., 2017). Such gaseous air pollutants are rapidly transformed into secondary organic

aerosol (SOA), for which the OS has been shown to be a large source (Liggio et al., 2016). Despite the large SOA formation rates observed in the OS (~45-84 ton day$^{-1}$) (Liggio et al., 2016), the emission sources, chemical compositions, volatilities and SOA forming potentials of the precursors remain unclear. Understanding the impact of SOA on the regional PM$_{2.5}$ burden, air quality and potentially regional climate requires accurate model predictions of SOA, which have been limited by the lack of data on OS source specific SOA precursors and their SOA forming potential (Stroud et al., 2018).

Investigating SOA forming potentials of hydrocarbons is generally accomplished through targeted experiments of single precursor compounds of interest to derive a yield (Odum et al., 1996;Kroll and Seinfeld, 2008). However, in the OS, SOA precursors are highly complex mixtures with volatilities spanning the range of volatile organic compounds (VOC; saturation concentration C* > 3×10$^6$ μg m$^{-3}$) to semi-volatile organic compounds (SVOC; C* = 0.3 – 300 μg m$^{-3}$) (Donahue et al., 2012;Liggio et al., 2016;Tokarek et al., 2018), hence making a single species approach to studying SOA formation

impractical. In addition, the mix of precursors (and hence chemical properties) varies by source within any given OS facility. Precursor emissions occur throughout the OS surface mining and processing production cycle, including sources such as open pit surface mines, processing plants and tailings ponds. The organic gases evaporated from these OS sources are mainly alkanes of diverse structure (e.g., linear, branched and cyclic) (Simpson et al., 2010;Li et al., 2017), which primarily react with OH radicals in the atmosphere, as their reactions with NO$_3$ radical and O$_3$ are very slow (Atkinson and Arey, 2003).

Consequently, the mixture of vapors evolved from the above sources are ideally suited to experimental studies of their overall SOA forming potentials/yields with oxidation flow reactors (OFRs) where ozone is often utilized as an OH radical precursor.

The development of OFRs has recently provided a complimentary approach to traditional large volume smog chambers to investigate SOA formation processes (Kang et al., 2007;Lambe et al., 2011;Bruns et al., 2015). The advantages associated

with the use of OFRs include their ability to probe the SOA forming potentials of a real-world mixture of precursors, and to study SOA formation on short time scales, simulating up to several weeks of OH radical exposure (Lambe et al., 2015;Bruns et al., 2015;Palm et al., 2016). OFRs have been utilized in numerous studies to investigate the SOA forming potentials of bulk gasoline and diesel emissions (Tkacik et al., 2014;Karjalainen et al., 2016;Jathar et al., 2017;Simonen et al., 2017),





biomass burning emissions (Ortega et al., 2013;Bruns et al., 2015), ambient air at numerous locations (Ortega et al., 2016;Palm et al., 2016) and single precursors (Kang et al., 2011;Lambe et al., 2011;Lambe et al., 2012;Lambe et al., 2015). The results of several OFR studies have also been used to infer the presence of gas-phase fragmentation reactions (i.e., C-C bond cleavage) (Jimenez et al., 2009), the transition between functionalization (i.e., oxygen addition) and fragmentation

(Lambe et al., 2012), and the corresponding impact of these processes on SOA formation yields of single precursors and complex mixtures (Lambe et al., 2012;Tkacik et al., 2014). However, results from OFR studies vary; with some single precursor experiments noting significantly lower SOA yields from OFRs compared to smog chambers (e.g., isoprene and *m*-xylene) (Lambe et al., 2011;Lambe et al., 2015), and others indicating similar but consistently lower yields than traditional smog chamber results (e.g., α-pinene) (Bruns et al., 2015;Lambe et al., 2015). Additionally, studies of vehicle exhaust

mixtures in OFRs generally exhibit reduced SOA potential relative to smog chambers at similar photochemical ages (Tkacik et al., 2014;Jathar et al., 2017;Simonen et al., 2017). Similarly, the impact of fragmentation on SOA yields in OFRs have been reported to be relatively large at moderate to high OH exposures in some studies (Lambe et al., 2012;Lambe et al., 2015) but negligible in others (Bruns et al., 2015). Although the use of OFRs has been suggested as a complimentary approach to smog chambers, such disparities between OFR experiments, and between OFR and chamber results, is likely to

make the interpretation of OFR SOA yields and their application to air quality modeling systems difficult. This is particularly relevant for the use of OFRs with a complex mixture of precursors.

In this study, the application of a newly developed OFR (the Environment and Climate Change Canada OFR, ECCC-OFR) to single compounds and complex precursor mixtures is described. The derived SOA yields for single compounds (alkanes and α-pinene) are compared with those of other OFRs and smog chambers, providing improved confidence in the use of

20 OFRs for the determination of SOA yields, and in the understanding of the relative importance of fragmentation processes to SOA formation. The ECCC-OFR is further used here to study the OH initiated formation of SOA from various OS derived complex mixtures under low-$NO_x$ conditions. These mixtures are representative of the potential pollution from distinct stages of the OS production cycle and/or sources. This new information on the yields of SOA from these varied OS sources and other single compounds will improve the understanding of SOA formation from this large industrial sector, advance the

25 modeling of the OS SOA formation in regional air quality models, and improve the overall understanding of alkane derived SOA in general.

## 2 Methods

SOA formation was investigated using a custom-made OFR (ECCC-OFR), which is shown schematically in Fig. S1. The design of the ECCC-OFR was partially based upon recent OFRs designs (Lambe et al., 2011;Huang et al., 2017;Simonen et

al., 2017) with several specific differences described further in the Supporting Information (Sect. S1). Briefly, the reactor is a fused quartz cylinder with a cone shaped diffusion inlet. The length of the cylinder is 50.8 cm, with an inner diameter of 20.3 cm. The length of the cone inlet is 35.6 cm, with a cone angle of 30˚. The conical inlet is designed to minimize the



establishment of jetting and recirculation in the OFR (Huang et al., 2017), which were noted for straight OFR inlets (Huang et al., 2017;Mitroo et al., 2018). There are seven openings at the output end of the ECCC-OFR; six of them (I.D.=0.25") are equally spaced around the perimeter to provide side flow as exhaust with a distance to the walls of 2.5 cm, intended to reduce the impact of gas and particle wall losses on sampling. A stainless steel sampling port (O.D.=0.25", I.D.=0.18") is

5 located in the center of the reactor, protruding 12.7 cm into the ECCC-OFR to minimize the influence of any potential turbulent eddies induced at the end of the reactor. While explicit fluid modeling was not conducted for this OFR, modeling of the flow in similarly designed OFRs indicates that a reasonable laminar flow is likely achieved (Huang et al., 2017). The volume from the inlet of the cone to the sampling port is approximately 16 L. The total flow rate for experiments is 8 L min$^{-1}$, resulting in a residence time of 120 s. The sampling flow rate is approximately 1.6 L min$^{-1}$ (determined by the flow of

10 instruments connected), with an additional flow (6.4 L min$^{-1}$) pushed out of the reactor through the side ports as exhaust. Four ozone-free Hg UV lamps (BHK Inc.) used to generate OH radical are located in four open-ended fused quartz tubes that are parallel and external to the OFR (1.5 cm). The lamps are purged by a large flow of air (~30 L min$^{-1}$) through the quartz tubes to remove the heat generated by lamps, resulting in a working temperature of approximately 25 ˚C, which is slightly higher than room temperature. The entire reactor is contained in an internally mirrored polycarbonate enclosure to

15 direct all produced light towards the reactor.

OH radicals were generated by photolysis of $O_3$ at 254 nm followed by reaction with water vapor, a commonly used method in many OFRs (Kang et al., 2007;Lambe et al., 2012;Liu et al., 2014). The relative humidity was detected at the outlet (side flow) of the reactor with a humidity sensor (Vaisala), and was maintained at 37% ± 2% at room temperature (21 ± 1˚C) by controlling the flow of dry and wet zero air into the reactor. The OH exposure (photochemical age) inside the reactor was

20 estimated through the decay of CO due to its reaction with OH (Li et al., 2015). The CO was introduced into the ECCC-OFR during separate experiments to characterize OH exposure off-line. The CO concentration was measured with a CO analyzer (LGR CO-23r) with a high precision (0.1 ppb). A low initial concentration of CO (~0.5 ppm) was used to minimize the external OH reactivity introduced by CO, hence increasing the accuracy of OH exposure estimation (Li et al., 2015). The OH radical concentration was adjusted through changes in the UV light intensity by varying the voltage applied to the lamps.

The OH exposure during experiments ranged from $1.2 \times 10^{10} - 2.3 \times 10^{12}$ molec cm$^{-3}$ s, which corresponds to 0.1 – 18 days of photochemical age, assuming a global average OH concentration of $1.5 \times 10^6$ molec cm$^{-3}$ (Mao et al., 2009). However, the equivalent aging time is significantly shorter for urban areas and OS production regions, because of their typically higher ambient OH concentrations (Hofzumahaus et al., 2009;Stone et al., 2012). For example, the OH exposure range is equivalent to 20 min – 2.7 days if assuming an average OH concentration of $1 \times 10^7$ molecules cm$^{-3}$ as has been estimated for the plumes

originating from Alberta OS operations (Liggio et al., 2016).

Vapors from α-pinene, individual alkanes (*n*-heptane, *n*-decane, *n*-dodecane, cyclodecane and decalin) and various OS related samples (with the exception of the tailings pond sample) were introduced into the ECCC-OFR by a small flow of zero air (0.5-2 ml min$^{-1}$) passing over the headspace of the sample material, which was placed in a glass U-tube and maintained at room temperature. The OS samples included freshly mined OS ore (original, unprocessed), bitumen (processed





heavy oil product), naphtha (a solvent used in OS processing), diluted bitumen (dilbit, a mixture of bitumen and solvent for transport in pipelines), and tailings pond water (waste water from the mining and processing) (see Supporting Information for details). The tailings pond sample (~2 L) was placed into a 4 L glass bottle and was bubbled into the ECCC-OFR. For some samples with high volatilities (e.g., naphtha and $n$-heptane), the gas-phase was further diluted before being injected

into the reactor. Total hydrocarbon concentration entering the ECCC-OFR was determined by passing the input gas stream (in off-line experiments) through a Pt based catalytic converter maintained at 400 ˚C and measuring the subsequently evolved $CO_2$ (Li-Cor LI-840A) as described by Veres et al. (2010). The evolved $CO_2$ concentration (ppb) is converted to the total C concentration (ppbC, see Table 1). The conversion efficiency of this total hydrocarbon (THC) system was measured to be 100±2% for several hydrocarbons in the range of $C_7$-$C_{18}$ (see Supporting Information and Fig. S6), but has been shown

to be equally efficient at lower carbon numbers (Veres et al., 2010). For complex OS precursor mixtures introduced into the OFR, a volatility distribution (VD) was measured by collecting the vapor-phase onto desorption tubes containing Tenax (Gerstel) followed by analysis with a thermo-desorption gas chromatography–mass spectrometer (TD-GC-MS, Agilent) utilizing a method described previously (Liggio et al., 2016).

Particle size distributions at the exit of the OFR were measured with a scanning mobility particle sizer (SMPS, TSI), which

were used to determine SOA yields. For a subset of experiments, ammonium sulfate (AS) seed particles were generated with an atomizer, dried with a diffusion dryer, and introduced into the reactor without size selecting. The mass concentration of the AS seed particles was approximately 20 μg m$^{-3}$ for most experiments with a number-weighted mode diameter of approximately 50 nm (mass-weighted mode diameter ~90 nm). For OS ore and naphtha, additional seed concentration experiments (~10 and 40 μg m$^{-3}$) were also performed to investigate the impact of seed concentration on SOA formation.

Particle composition was determined using a long time-of-flight aerosol mass spectrometer (LToF-AMS, Aerodyne) with a mass resolution of 6000-8000 in V-mode. The mass spectra and elemental properties of the SOA were determined using the AMS analysis software Squirrel (Version 1.57I) and Pika (Version 1.16I). The elemental ratios (H/C and O/C) were estimated using the improved ambient method described previously (Canagaratna et al., 2015). The SOA mass concentration was calculated by multiplying the integrated volume concentration from the SMPS (after subtracting the AS volume

concentration for seeded experiments) by the effective particle density. The effective density (ρ, 1.35-1.6 for different precursors) was calculated from the vacuum aerodynamic diameter ($D_{va}$, obtained from the AMS) and the electric mobility diameter ($D_m$, obtained from the SMPS) for non-seeded experiments using the equation $\rho = D_{va} / D_m$ (Lambe et al., 2015). The same density was used for seeded and non-seeded experiments.

In the current study, only low-NO$_x$ experiments were performed for all precursors, in which the reaction with HO$_2$ radical

dominates the fate of the peroxy radical (RO$_2$) formed in the initial OH reaction. Such conditions are likely to represent the atmospherically relevant scenario where OS emissions have been transported significantly downwind of the OS region (and NO consumed), over boreal forest areas, where there were few NO$_x$ sources. In addition, the low-NO$_x$ condition is a typical oxidation pathway parameterized in regional air quality models. The formation of OS derived SOA under high-NO$_x$ conditions (closer to sources) is the topic of a forthcoming publication.



## 3 Results and discussion

### 3.1 Characterization of the ECCC-OFR

#### 3.1.1 Wall losses

Previous OFR studies have indicated that the wall losses of both gaseous precursors and formed particles are potentially
important factors in influencing the SOA yield results from OFRs (Lambe et al., 2011;Lambe et al., 2015;Huang et al.,
2017;Simonen et al., 2017). The particle wall losses for the ECCC-OFR were assessed by measuring size-selected (50 nm,
100 nm, 150 nm and 200 nm diameter) ammonium sulfate (AS) (Huang et al., 2017) and bis(2-ethylhexyl) sebacate (BES)
(Lambe et al., 2011;Simonen et al., 2017) aerosol number concentrations before entering and after exiting the reactor. As
shown in Fig. 1, the concentration of AS aerosols after the reactor is within ±3% of the concentration before the reactor. For
BES, the particle transmission efficiency ($P_{trans}$) is 92% at 50 nm and increases to ~100% for 100 nm and larger particles.
This indicates that particle wall losses of the ECCC-OFR were very small for the flow conditions and particle size range in
the experiments, and hence were not considered in further SOA yield calculations. The $P_{trans}$ of other OFRs are also shown in
Fig. 1 for comparison and indicates that the current $P_{trans}$ is similar to that of the TSAR (TUT Secondary Aerosol Reactor)
(Simonen et al., 2017), likely due to the similarity in design (i.e., cone shaped inlet and sampling from the center-line, see
Supporting Information). Conversely, the $P_{trans}$ of the TPOT (Toronto Photo-Oxidation Tube), PAM (Potential Aerosol
Mass) reactor (Lambe et al., 2011) and CPOT (Caltech Photooxidation Flow Tube) (Huang et al., 2017) are 15-85%, 20-60%
and 20-45% lower respectively than the ECCC-OFR across a range of particle sizes. Potential causes of these discrepancies
include recirculation and turbulence induced by a straight inlet and/or output sampling end (Lambe et al., 2011), non-
centerline sampling (Huang et al., 2017) and longer residence times (Huang et al., 2017) in the other OFRs (see Supporting
Information), which have been noted as potential factors previously (Lambe et al., 2011;Simonen et al., 2017;Mitroo et al.,
2018).

The transmission efficiencies of the ECCC-OFR for gaseous hydrocarbons ($G_{trans}$) in the volatile to intermediate volatility
ranges were also determined using the THC conversion methodology described above to measure the concentration
immediately before entering and after exiting the reactor. The $G_{trans}$ results for three $n$-alkanes, specifically $n$-heptane ($C_7$), $n$-
decane ($C_{10}$) and $n$-dodecane ($C_{12}$) are shown in Fig. 1, and are approximately 100%±3%. Measurement data with respect to
hydrocarbon transmission efficiency for the other OFRs are not currently available for comparison. While the loss of
hydrocarbon precursors in the ECCC-OFR may be minimal, one cannot easily measure the losses of lower volatility
oxygenated compounds directly, particularly those of intermediate products of oxidation, which largely influence measured
SOA yields in smog chambers and the other OFRs (Zhang et al., 2014;Palm et al., 2016). However, an approximate
indication of the potential for gaseous wall losses is provided by estimating the characteristic time associated with diffusion
from the center to walls of the ECCC-OFR as described previously (Huang et al., 2017). The characteristic diffusion time
($\tau_c$) in the radial direction is given by



$$\tau_c = \frac{R^2}{D_i},\tag{1}$$

where R is the inner radius of the ECCC-OFR, and $D_i$ is the molecular diffusivity of species i in air (Huang et al., 2017). A typical estimate of the molecular diffusivity for oxidized organic vapor (with a molecular weight of 200 g mol$^{-1}$) in air is ~$7\times10^{-6}$ m$^2$ s$^{-1}$ (Tang et al., 2015;Palm et al., 2016), leading to a $\tau_c$ of ~1400 s. This characteristic diffusion time is much

longer than the average residence time in the reactor (120 s), suggesting that the interaction of gases at the ECCC-OFR center with the walls is likely very small. As a result, we suggest that the wall losses for intermediate oxygenated products on the measured SOA yields were minor, although this would in part also depend on the ideality of laminar flow at the sampling point of the ECCC-OFR which was not assessed.

### 3.1.2 SOA yields and fragmentation

An important performance characteristic of an OFR is the ability to derive SOA yields consistent with previous results in traditional chamber experiments (Bruns et al., 2015;Lambe et al., 2015). The SOA yields from the ECCC-OFR (under low-NO$_x$ conditions), for selected individual compounds (α-pinene, *n*-decane (C$_{10}$), *n*-dodecane (C$_{12}$)), as a function of photochemical age or OH exposure and in the presence or absence of AS seed aerosol are provided in Fig. 2. The SOA yields (Y) in Fig. 2 are calculated using the mass concentration of organic aerosols ($\Delta M_O$) and reacted parent hydrocarbons

($\Delta HC$, see Supporting Information for details), where Y = $\Delta M_O$ / $\Delta HC$. Figure 2 also shows the yields from other recent smog chamber and OFR studies for the same individual precursors under low-NO$_x$ conditions (see Table 2 for details) (Ng et al., 2007;Eddingsaas et al., 2012;Lambe et al., 2012;Chen et al., 2013;Loza et al., 2014;Lambe et al., 2015;Bruns et al., 2015;Han et al., 2016).

As most previous smog chamber studies are carried out at relatively low OH exposures, limited data can be used for

comparison, since the majority of chamber data resides in the photochemical age less than 3 equivalent days ($3.9 \times 10^{11}$ molecules OH cm$^{-3}$ s$^{-1}$, Table 2). However, in addition to the OH exposure level, numerous other factors may affect the SOA yield comparisons between OFR and chambers. These factors include the concentration of gas phase precursor utilized, the presence or absence of seed aerosol, and the mass of SOA formed during experiments (Odum et al., 1996;Donahue et al., 2006;Kroll et al., 2007;Kroll and Seinfeld, 2008;Hallquist et al., 2009). Nonetheless, the α-pinene SOA yields in the ECCC-

OFR are similar to previous chamber experiments at similar OH exposures (Fig. 2a, Table 2). Given the known dependence of yield on SOA mass and precursor concentration (Odum et al., 1996;Kroll and Seinfeld, 2008), slightly higher yields for α-pinene are expected from chamber studies (and observed), as some experiments were performed at SOA mass levels and gaseous precursor concentrations 3-14 and 3-15 times (Ng et al., 2007;Eddingsaas et al., 2012;Bruns et al., 2015) greater than the current study (22-42 μg m$^{-3}$ and 13.7 ppb; see Table 2 for details). Considering the impact of these conditions on

yields, the ECCC-OFR SOA yields of α-pinene are in reasonable agreement with those derived from chamber studies. However, in the case of alkanes, the agreement is significantly different. While the initial *n*-dodecane concentration and OA concentration (upper limit) in a previous study (Loza et al., 2014) were ~3 times higher than this study (Table 2), the



corresponding SOA yields were significantly lower (Fig. 2b) than the current results. The known impact of gaseous wall loses on SOA yields in environmental chambers (Zhang et al., 2014) suggests the long residence time of those particular experiments (~36 hours) (Loza et al., 2014) likely resulted in significant intermediate gaseous product wall losses and correspondingly low SOA yields compared to the ECCC-OFR (which has minimal wall losses).

While the SOA yields for single precursors from the present study are in reasonable agreement with traditional chamber data, they are significantly larger than those of other OFR data sets (Lambe et al., 2012;Chen et al., 2013;Lambe et al., 2015;Bruns et al., 2015) (Figs. 2a and b). With the exception of the lowest OH exposure data point for α-pinene oxidation, the SOA yields quickly diverge from each other after approximately 2 equivalent photochemical days (a factor of 4 larger after ~10 equivalent days) for unseeded experiments. This despite initial concentrations of α-pinene (41-100 ppb) and SOA

mass (90 μg m$^{-3}$) in previous OFR experiments (Chen et al., 2013;Lambe et al., 2015) being considerably higher than the current study (13.3 ppb and 37.9 μg m$^{-3}$) at similar photochemical ages (Table 2). For seeded experiments of α-pinene, the current SOA yields are higher than those reported by Bruns et al. (2015), despite their precursor concentration and SOA mass being 10-25 and 5-24 times higher than this study (13.7 ppb and 41.9 μg m$^{-3}$, Table 2). Similarly, the present SOA yields for $n$-decane ($C_{10}$) diverge from previously reported results (Lambe et al., 2012) (Fig. 2b), with the present SOA yields

up to a factor of 4 higher after ~10 equivalent photochemical days ($1.3 \times 10^{12}$ molecules cm$^{-3}$ s OH exposure). It is noteworthy that the yields for $n$-decane from the present study and reported by Lambe et al. (2012) are in reasonable agreement for up to 2 equivalent days ($2.6 \times 10^{11}$ molecules cm$^{-3}$ s OH exposure). However, this is likely fortuitous, as the SOA mass concentration and precursor concentration in the study by Lambe et al. (2012) (231 μg m$^{-3}$ and 102 ppb) was an order of magnitude higher than in the present study (30.4 μg m$^{-3}$ and 23.4 ppb, Table 2), which will enhance the gas-particle

partitioning process and lead to higher yields. Such an effect has been observed in $C_{15}$ SOA experiments (Lambe et al., 2012), where decreasing the aerosol mass concentration from 100 μg m$^{-3}$ to 16 μg m$^{-3}$ reduced the SOA yield from 0.69 to 0.21.

The decrease in yield at longer photochemical ages (higher OH exposures) in previous OFR studies (Figs. 2a and b) has been attributed to gas-phase fragmentation leading to higher volatility SOA products, with a transition point between

functionalization and fragmentation observed at the maximum carbon yield (Lambe et al., 2012). The SOA carbon and oxygen yields ($Y_C$ and $Y_O$) for α-pinene and $n$-decane from the current experiments are shown in Fig. 2c following the approach outlined elsewhere (Kroll et al., 2009;Lambe et al., 2012) and presented in details in the Supporting Information. In the absence of gaseous wall losses, the impact of fragmentation may be indicated by a relatively larger decrease in $Y_C$ at higher OH exposure compared to $Y_O$ (Kroll et al., 2009;Lambe et al., 2012). Such an effect is observed in the present results

for both α-pinene and n-decane (Fig. 2c), with $Y_C$ decreasing by 38% and 15% over 7 and 13 photochemical days, respectively. The maximum $Y_O$ is at a higher photochemical age compared to $Y_C$ for SOA formed from both precursors (~9 and 4 photochemical days for α-pinene; ~13 and 6 photochemical days for $n$-decane), further consistent with a transition from functionalization to fragmentation in these experiments as indicated in Fig. 2c. However, the relative impact of fragmentation on the overall SOA yields here is in contrast to that suggested previously (Lambe et al., 2012) (Figs. 2a and



b). The maximum $Y_C$ for *n*-decane here is observed at a higher photochemical age of 6 days, compared to 4 days seen by Lambe et al. (2012), and the decrease in $Y_C$ and overall Y is also significantly less (15% vs ~95% for $Y_C$; <5% vs ~95% for Y).

Given the similarity in the OH exposure range used between studies, and the generally higher SOA mass concentration (and precursor concentration) in previous OFR studies (Lambe et al., 2012;Chen et al., 2013;Lambe et al., 2015), the present results suggest that gaseous wall losses during the oxidation process may have reduced previously observed yields in their OFRs, thereby leading to an overemphasis on the importance of fragmentation to SOA formation. It is notable that the relative impact of fragmentation here, although small, may also not be fully applicable to the ambient atmosphere due to the fate of low volatility organic compounds (LVOCs) in the OFRs. Accounting for the fate of LVOCs reduces the potential importance of fragmentation to SOA formation in this study and the ambient atmosphere even further, as is described below (Sect. 3.1.3).

### 3.1.3 Fate of LVOCs

Previous studies have demonstrated that SOA yields derived in OFRs at high OH exposures (and other conditions) have likely been underestimated, due to differences between the fates of LVOCs in OFRs and the ambient atmosphere (Palm et al., 2016). There are four possible fates associated with LVOCs in an OFR: condensation to aerosol, reaction with OH, condensation to the OFR walls, and exiting the OFR (then lost on sampling walls). However, in the ambient atmosphere, condensation to aerosol is the dominant fate of LVOCs, indicating that the other three possible fates are limitations of the OFR (Palm et al., 2016). To characterize the ECCC-OFR with respect to the fate of LVOC and improve the subsequent applicability of the data to the ambient atmosphere, we modeled the fate of LVOCs under conditions specific to these experiments, following the approach of Palm et al. (2016), as described further in the Supporting Information.

The modeled fates of LVOCs in the ECCC-OFR for unseeded and AS seeded conditions are shown in Figs. 3a and b, using the parameters (OH concentration and aerosol size distribution) from α-pinene experiments. Figure 3a indicates that condensation on aerosol surfaces (in the absence of seed particles, for α-pinene derived SOA) accounts for 70%-80% of the LVOC fate between ~1-6 photochemical days, decreasing to 40%-50% at 16 photochemical days. These fractions are similar to an ambient OFR study conducted in Los Angeles (~40%-80%) (Ortega et al., 2016), but higher than that obtained at a forested site (~10%-70%) (Palm et al., 2016). OH oxidation accounts for 5%-50% of the LVOC loss in the ECCC-OFR, increasing in importance at higher photochemical age, while LVOC wall losses and OFR exiting fates are very small, generally less than 5%. It should be noted that the wall loss fate of LVOC is calculated based upon the surface-area-to-volume ratio of the OFR, without considering the effect of centerline sampling, which would make the wall loss fate contribution even smaller. For experiments using 20 μg m$^{-3}$ of AS seed particles, the fraction that condenses onto aerosol (~70-95%; Fig. 3b) is significantly higher than that for unseeded experiments, due to the presence of a higher condensational sink.



The fraction of LVOCs that condenses on aerosol ($F_{aerosol}$) for single precursors ($\alpha$-pinene, *n*-decane, and *n*-dodecane) and various OS-related precursor mixtures (their yields will be discussed in the following section) is shown in Figs. 3c and d. The $F_{aerosol}$ are very similar to each other in the presence of AS seed particles regardless of the precursor, accounting for ~95% of the LVOC fate at less than 1 photochemical day and ~70% at ~16 photochemical days. Conversely, the range of

$F_{aerosol}$ is much wider for non-seeded experiments (Figs. 3c and d), from ~40% to 80%. The results suggest that the OFR experiments under the seeded conditions here are the most relevant to the ambient atmosphere, particularly at less than 4 photochemical days, with yields potentially requiring a relatively small upwards adjustment (~30%) even at >14 photochemical days. The model also suggests that the impact of fragmentation reactions on SOA yields (derived from this OFR), when translated to the atmosphere, is likely to be very small, as the OH reactions of LVOC never dominate the overall

fate (Fig. 3b).

Given the results of Fig. 3, future OFR studies investigating SOA yields should be conducted in the presence of pre-existing seed particles to reduce uncertainties, as theoretically suggested previously (Palm et al., 2016). The estimated fate of LVOCs for seeded experiments here is used to apply an upwards correction to $\alpha$-pinene (Fig. S4) and OS derived SOA yields (discussed in Sect. 3.2) assuming an LVOCs fraction of 80% in SOA (see Sect. S5 of Supporting Information for details). As

OH concentrations in smog chambers are generally much lower than studies with the OFRs, the LVOCs in smog chamber will mostly condense on aerosols, which is similar to the real atmosphere. Hence, when comparing the OFR yields to smog chambers, an LVOC fate correction should be applied. As shown in Fig. S4, the SOA yields from $\alpha$-pinene in the current OFR after correction are in good agreement with previous smog chamber results despite the lower SOA mass concentration and precursor concentration.

## 3.2 SOA yields of OS-related precursors

The ECCC-OFR was used to investigate the SOA yields of complex precursor mixtures; specifically those derived from OS sources (see Methods). The SOA yields of these OS-related precursor mixtures are shown in Fig. 4a for unseeded experiments performed in an atmospherically relevant SOA mass concentration range (< 50 µg m$^{-3}$; Table 1). The SOA yield in this case is defined similarly to that in Sect. 3.1.2, but accounting for the calculated H/C ratio (Table S1) and measured

carbon number distribution of emissions (Fig. 5a), as described in detail in the Supporting Information. Briefly, the H/C ratios of precursors were used to calculate the initial precursor mass concentrations from the measured total carbon concentration. The reacted mass concentrations were calculated using the rate constant with OH of corresponding *n*-alkanes that have the same carbon number as the average value of carbon number distributions. As demonstrated in Fig. 4a, the freshly mined OS ore results in the highest yields among the five precursor mixtures, with a maximum of 0.44±0.05 at

approximately 11 atmospheric equivalent photochemical days ($1.4 \times 10^{12}$ molecules cm$^{-3}$ s OH exposure, corresponding to approximately 1.6 days in OS plumes (Liggio et al., 2016)), followed by processed bitumen, with slightly lower yields over the entire range of photochemical age (with a maximum of 0.35±0.03). The SOA yields of naphtha, dilbit and tailings pond emissions are significantly lower, with maximum SOA yields of approximately 0.1±0.01 to 0.13±0.01. The difference in



yields between source mixtures (Fig. 4a) can be qualitatively explained by the volatility distributions (VD) of these precursors (Fig. 5), with precursors of lower volatility (higher carbon number) having higher SOA yields (Lim and Ziemann, 2005;Lim and Ziemann, 2009). In this case, naphtha solvent and OS ore emissions represent volatility endpoints (high and low respectively) with other precursor mixtures being derived from a combination of these (see Sect. S3 of Supporting Information for details).

SOA yields from several straight chain pure compounds ($C_7$, $C_{10}$, and $C_{12}$) were also investigated in the ECCC-OFR to provide additional information on the nature of the OS related precursor mixtures, and are depicted in Fig. 4a. These single compounds were selected for comparison based upon the VD of the OS precursors (Fig. 5a), where heptane ($C_7$) represents the maximum of the VD of naphtha and dilbit, decane ($C_{10}$) the approximate average volatility of OS ore (see Sect. S3 of Supporting Information) and dodecane ($C_{12}$) a compound representative of the lower end of the VD of OS ore and processed bitumen. As shown in Fig. 4a, despite naphtha and dilbit vapors being dominated by compounds with an equivalent volatility to heptane (Fig. 5a), their SOA yields ($0.11 \pm 0.01$) are significantly higher than that of heptane ($0.044 \pm 0.006$). Similarly, OS ore emissions result in higher yields than decane, despite a comparable volatility, but lower yields than $C_{12}$. This suggests that alkanes with higher carbon number (and hence lower volatility and higher yield) contribute disproportionately to the overall SOA yields, relative to their proportions in the precursor emissions (Fig. 5a). Alternatively, cyclic hydrocarbons in the OS-related precursors could also contribute significantly to the overall yields, as experiments for cyclodecane and decalin (a bicyclic $C_{10}$ alkane) (Fig. 4a) result in much higher yields than decane. This is consistent with previous studies that demonstrated that cyclic alkanes have much higher yields than *n*-alkanes in general (Lim and Ziemann, 2009;Tkacik et al., 2012;Hunter et al., 2014). While the yields for single species alone cannot be used to distinguish between the contributions of cyclic and acyclic compounds to the observed OS derived SOA, elemental ratios of the SOA suggest that cyclic species may be an important contributor (see Sect. 3.3).

The SOA carbon and oxygen yields ($Y_C$ and $Y_O$) for the least and most volatile precursor mixtures (OS ore and naphtha solvent respectively) are shown in Fig. 4c, as an indicator of the impact of fragmentation on the derived SOA yields. Both $Y_C$ and $Y_O$ for OS ore and naphtha reach a maximum at approximately 11 equivalent photochemical days, and then decrease with increasing photochemical age. The decrease in $Y_O$ for OS ore and naphtha is ~1% per equivalent day from 11 to ~15-17 days. However, the $Y_C$ for OS ore and naphtha decrease ~2%-4% per day, which is higher than the decrease in $Y_O$. This suggests that fragmentation reactions increasingly influence SOA yields at higher photochemical ages for OS-related precursors, although a significant relationship between the degree of fragmentation and carbon number cannot be determined. Regardless, the overall impact of the competition between functionalization and fragmentation on the SOA yields here is small across all OS derived precursors. This is in contrast to other types of fuel products, specifically diesel and Southern Louisiana crude oil (Fig. 4c), which were shown to have SOA yields that are highly affected by fragmentation reactions (Lambe et al., 2012), although those studies were likely impacted by wall losses.

The results of experiments conducted using 20 µg m$^{-3}$ solid ammonium sulfate (AS) seed particles are shown in Fig. 4b. Experiments with 10 and 40 µg m$^{-3}$ AS seed particles were also performed for OS ore and naphtha, but exhibited no SOA




yield dependence on seed concentration (not shown), with the same SOA yields derived in all cases. Generally, the SOA yields for all precursors are enhanced significantly in the presence of AS seed particles, with maximum yields of 0.58±0.03 and 0.18±0.02 for the least and most volatile OS precursors. This effect is more clearly depicted as a yield enhancement ratio ($Y_{seeded}/Y_{unseeded}$) in Fig. 4d. Based upon Fig. 4d, it is evident that SOA from precursors with higher volatilities are more

5  impacted by the presence of AS seed particles; SOA yield enhancement ratios for naphtha and dilbit (~60%) are higher than OS ore and bitumen (~30%) after approximately 2 equivalent photochemical days, with that of tailings pond SOA between them. It is also evident that the enhancement factor is somewhat larger during the initial stages of oxidation (up to >100% at <2 equivalent photochemical days). This is likely a result of the different LVOCs fate for seeded and unseeded experiments. As shown in Sect. 3.1.3 and Fig. 3d, the fraction of LVOCs that condense on aerosol ($F_{aerosol}$) at <2 equivalent photochemical

10  days for unseeded experiments is much lower than that for seeded experiments, which will lead to a larger yield enhancement ratio in the presence of seed particles. The finding that the presence of seeds can enhance the SOA yields is in agreement with various previous work (Kroll et al., 2007;Hildebrandt et al., 2009;Zhang et al., 2014;Lambe et al., 2015;Li et al., 2018). In addition to the difference in LVOCs fate discussed above, the enhanced SOA yield in the presence of seed particles can also be due to increased aerosol surface area that competes with other sinks (e.g., vapor wall losses for smog

chambers) and enhances the gas-particle partitioning of S/IVOCs, as suggested previously (Hildebrandt et al., 2009;Zhang et al., 2014;Li et al., 2018).

The OS precursor SOA yields for seeded experiments are adjusted upwards to account for the fate of formed LVOCs through normalization by the $F_{aerosol}$ above (Sect. 3.1.3), with the results of this correction shown in Fig. 6a. Here, we assume 80% of the SOA is LVOCs, while the other 20% is S/IVOCs (see Sect. S5 of Supporting Information for details). Relative to

the yields of Fig. 4b, the LVOCs fate adjusted SOA yields of Fig. 6a are ~4% to 37% larger for all precursors, depending upon the OH exposure. As noted above, the fate of LVOC in seeded experiments is primarily condensation to the aerosols, requiring a relatively small adjustment. As a result, the seeded experiment data represent our best estimate of the SOA yields for the precursors in Fig. 4, applicable to the ambient atmosphere (under these conditions). In this case, the maximum SOA yield for the least and most volatile precursor mixtures (OS ore and naphtha) increased from 0.58±0.03 to 0.71±0.04 and

0.18±0.02 to 0.23±0.02 respectively after adjustment (Fig. 6a). In addition, applying an LVOC fate adjustment results in SOA yields for most OS precursors, α-pinene, and *n*-alkanes generally increasing with increasing OH exposure (compared to those of Figs. 2 and 4). This further suggests, as noted above, that the fragmentation reactions will not significantly decrease the SOA yields for these species in the ambient atmosphere, even after 16 equivalent photochemical days. However, uncertainties still remain when using OFRs to simulate the SOA formation processes in the real atmosphere, likely from

differing fates of intermediate radicals (e.g., $RO_2$) especially at high OH exposure as suggested very recently (Peng et al., 2019).



### 3.3 Elemental ratios of OS-related SOA

The elemental H/C and O/C ratios of SOA particles are illustrated in a Van Krevelen diagram (Heald et al., 2010) in Fig. 6b. Figure 6b indicates that the elemental ratios of SOA from OS ore and bitumen (and its photochemical evolution) are very similar (O/C: 0.45-0.8, H/C: 1.4-1.6), as are the elemental ratios of SOA formed from naphtha, dilbit and tailings pond water

(O/C; 0.6-0.9, H/C: 1.5-1.7). This is analogous to the similarity in the yields between the same precursors as discussed above (Fig. 4) and consistent with the volatility of the precursors (Fig. 5). The lower O/C ratios of OS ore and bitumen SOA is probably due to their larger molecular size, with higher carbon number (i.e., lower volatility) precursors requiring less oxygen (hence fewer oxidation steps) to partition into the particle phase (Tkacik et al., 2012). The H/C ratios are also lower for SOA formed from lower volatility precursor mixtures, which is likely a result of different H/C of the precursors, with

generally lower H/C for higher carbon number hydrocarbons. Assuming a linear relationship in Fig. 6b, the y-intercept is indicative of the average H/C of the precursor mixture (Fig. S5). The intercept of naphtha and dilbit SOA (~2.1) is higher than OS ore and bitumen SOA (~1.8), indicating a higher H/C ratio for those precursors.

Similar inferences are made when comparing the evolution of the elemental ratios of SOA from various single alkane species in Van Krevelen space to that of OS precursors (Fig. 6b). For example, SOA from parent $n$-alkanes with successively higher

carbon number (and lower volatility) move towards the bottom-left of the Van Krevelen diagram. However, the position of OS-related SOA in Van Krevelen space is not consistent with the corresponding $n$-alkanes; naphtha, dilbit and tailings SOA reside below $n$-heptane ($C_7$), despite having a very similar volatility (Fig. 5a). Similarly, OS ore and bitumen, reside below $n$-dodecane ($C_{12}$), despite $C_{12}$ volatility compounds contributing little to the overall volatility distribution of precursors (Fig. 5a). This discrepancy may be explained by the contribution of cyclic alkanes, since SOA formed from cyclic structures tend

to reside below acyclic alkane SOA in Van Krevelen space, and near that of OS derived SOA (e.g., cyclodecane and decalin relative to decane SOA and OS ore SOA in Fig. 6b). Recent aircraft measurement indicated that the cycloalkanes contribute 13%-27% of the total alkanes (Li et al., 2017) for Suncor and CNRL facilities (where the OS samples were collected), which will contribute a large proportion of SOA after considering their high SOA yields (Figs. 4a, b and 6a). A lower H/C ratio for SOA derived from cyclic alkanes is consistent with the parent hydrocarbon having lower H/C. The linear regression results

of H/C vs O/C for alkane precursors are listed in Table S1, from which the relationship between precursor H/C and intercept is obtained (see Sect. S4 in Supporting Information for details). A comparison between the H/C ratios of alkanes and OS precursors demonstrates that the H/C ratios of the OS precursors are generally lower than that of the corresponding $n$-alkane (e.g., ~2.2 for naphtha and dilbit, ~2.3 for $C_7$; ~2 for OS ore, ~2.2 for $C_{10}$), likely from the contribution of cyclic alkanes. Aromatics may also play a role in the decrease of H/C ratio of precursors; however, their contributions are likely small

according to recent aircraft measurement by Li et al. (2017) (e.g., 3.7% aromatics compared to alkanes for CNRL). In addition, the presence of aromatics will not decrease the observed H/C and O/C of SOA, for example, the H/C and O/C of toluene SOA (1.67 and 0.85) (Canagaratna et al., 2015) is similar to that of heptane SOA observed here. While the current



data cannot quantitatively apportion OS precursors to various structures (cyclic vs *n*-alkane/branched), the above Van Krevelen analysis suggests that cyclic compounds are an important contributor to the observed SOA.

The locations of two broad types of SOA, SV-OOA and LV-OOA (semi-volatile and low volatility oxidized organic aerosol) from various studies (Ng et al., 2011;Canagaratna et al., 2015), and that of the SOA downwind of the oil sands from previous aircraft measurements (Liggio et al., 2016) in Van Krevelen space are also shown in Fig. 6b. The positions of SOA formed from OS-related precursors in the ECCC-OFR are generally in the range of previous ambient OOA. They are in good agreement with SV-OOA and LV-OOA for experiments simulating ~2 photochemical days (~$2.6\times10^{11}$ molecules cm$^{-3}$ s OH exposure) and ~2 weeks (~$2\times10^{12}$ molecules cm$^{-3}$ s OH exposure), respectively. Furthermore, OS ore and bitumen derived SOA are more similar to ambient SV-OOA and LV-OOA than naphtha, dilbit and tailings pond water derived SOA (Fig. 6b). This highlights the contribution of intermediate-volatility alkanes to ambient SOA in the oil sands, particularly since the SOA formed from OS ore and bitumen are in good agreement with the aircraft data (Liggio et al., 2016) (Fig. 6b). Hence, these results indicate that low-volatility precursors from open-pit mining sources (i.e., OS ore) are likely the largest contributors to the SOA formed downwind of the Alberta OS region, while precursors of high volatility play a minor role, likely due to their lower SOA yields.

## 4 Conclusions and implications

In this study, a newly designed oxidation flow reactor (ECCC-OFR) was applied to the investigation of SOA formation from single precursor compounds (α-pinene, *n*-alkanes, and cyclic alkanes) and complex mixtures (OS-related precursors). The SOA yields for α-pinene and alkanes obtained in the ECCC-OFR are similar to previous smog chamber studies but significantly higher than other OFRs. The current results provide SOA yield information for alkane precursors for which limited data are available especially at moderate to high photochemical ages (Tkacik et al., 2012;Lambe et al., 2012). In addition, the differences in yields between the current and other OFRs suggests that while OFRs can provide insight into SOA mechanisms, care must be taken in deriving quantitative results from OFRs, which are often designed with slightly different geometries and operated under a variety of conditions. For example, recent OFR modeling results (Peng et al., 2019) demonstrated that the working conditions (e.g., light intensity and wavelength, humidity and external OH reactivity) could influence the RO$_2$ fate and result in less atmospherically relevant chemical mechanisms for SOA formation in the OFR.

Variability in the qualitative/mechanistic SOA information derived from OFRs is also possible. In particular, previous OFR studies (Lambe et al., 2012;Chen et al., 2013;Tkacik et al., 2014;Lambe et al., 2015;Ortega et al., 2016;Palm et al., 2016) have attributed large decreases in SOA yields at moderate to high photochemical age (typically after 4-5 equivalent days) to the dominant role of gas-phase fragmentation reactions. However, the current study indicates that the impact of fragmentation on SOA yields from various sources is minimal in the ECCC-OFR, likely due to reduced wall losses relative to other OFRs, whose fluid dynamics are not entirely laminar as suggested previously (Huang et al., 2017;Mitroo et al.,

2018). Accounting for the fate of LVOCs (Palm et al., 2016) in the ECCC-OFR further indicates that the impact of fragmentation on SOA yields in the ambient atmosphere will be even smaller than that within the OFR. This implies that modeling SOA formation to include the impacts of fragmentation should be carefully evaluated, especially if using OFR data to provide empirical factors for fragmentation (Chen et al., 2013). However, the current data also indicate that the impact of

fragmentation on SOA yields in OFRs can be significantly reduced through the use of seed particles, which increase the fraction of LVOCs which condense on aerosols ($F_{aerosol}$). This suggests that all future OFR experiments should be conducted with seed particles to obtain more relevant qualitative and quantitative data.

Application of the ECCC-OFR to OS-related precursor mixtures indicates that lower volatility OS ore and bitumen vapors have significantly higher yields (maximum of ~0.6-0.7 for seeded experiments after LVOCs fate correction) than those from

higher volatility naphtha, dilbit and tailings pond vapors (maximum of ~0.2-0.3 under the same conditions). The relatively high SOA yields from OS ore, together with the similar elemental ratios between ambient measurements and OFR experiments, is consistent with open-pit mining activities being the largest contributor to the observed SOA downwind of the oil sands operations (Liggio et al., 2016). The SOA yields and elemental ratio analysis also suggest that cyclic alkanes are import contributors to OS-related SOA. The OS SOA information derived here, for the range of precursor mixtures

encountered in the oil sands, can be used to improve parameterizations of SOA for the OS region through source specific inputs of SOA precursor properties and SOA yields, and to evaluate the subsequent regional modeling of SOA (Stroud et al., 2018). The attribution of observed industrial SOA in the oil sands to specific sources (i.e., OS ore emissions from open-pit mining) supports the potential for future mitigation strategies for reducing SOA from this sector.

**Author contribution**

KL and JL designed the OFR and the experiments; KL conducted the experiments; PL and KL measured the volatility distributions; KL and JL wrote the paper with contributions from all co-authors.

**Acknowledgements**

The authors acknowledge funding support from the Air Pollution program of Environment and Climate Change Canada (ECCC), and the Oil Sands Monitoring program (OSM). We further thank the Canada's Oil Sands Innovation Alliance

(COSIA) for the organization and provision of oil sands related samples used in this paper.

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





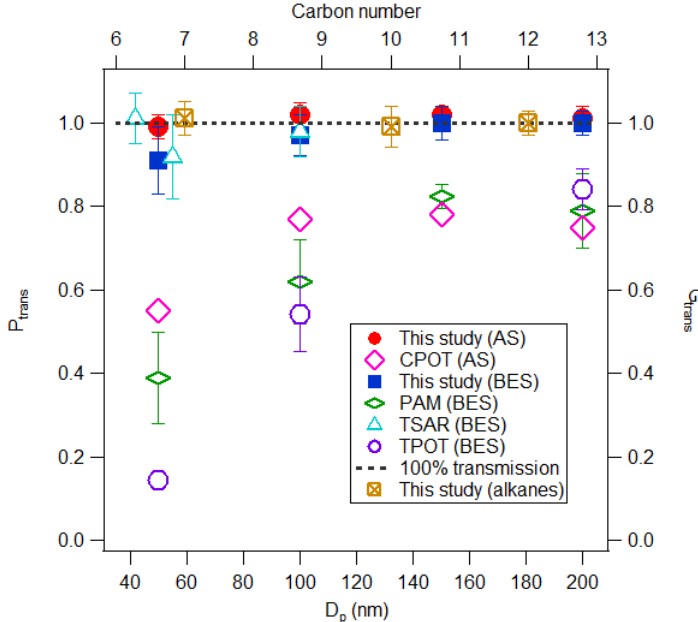

**Figure 1. Particle (left and bottom axis) and gas (right and top axis) transmission efficiencies (P*trans* and G*trans*) for the ECCC-OFR. Particle transmission efficiencies of other OFRs are shown for comparison: PAM (Lambe et al., 2011), TPOT (Lambe et al., 2011), TSAR (Simonen et al., 2017), and CPOT (Huang et al., 2017).**







**Figure 2. Low-NOₓ SOA yields of α-pinene (a), *n*-decane (C₁₀) and *n*-dodecane (C₁₂) (b), compared to previous studies using OFRs and smog chambers (SCs) (Ng et al., 2007;Eddingsaas et al., 2012;Lambe et al., 2012;Chen et al., 2013;Loza et al., 2014;Lambe et al., 2015;Bruns et al., 2015;Han et al., 2016). The details regarding these comparisons are shown in Table 2. (c): SOA carbon and oxygen yields (Y_C and Y_O) for single precursors for unseeded experiments in the current study and in a previous study (Lambe et al., 2012). Dashed and solid arrows indicate the maximum of Y_C and Y_O, respectively.**





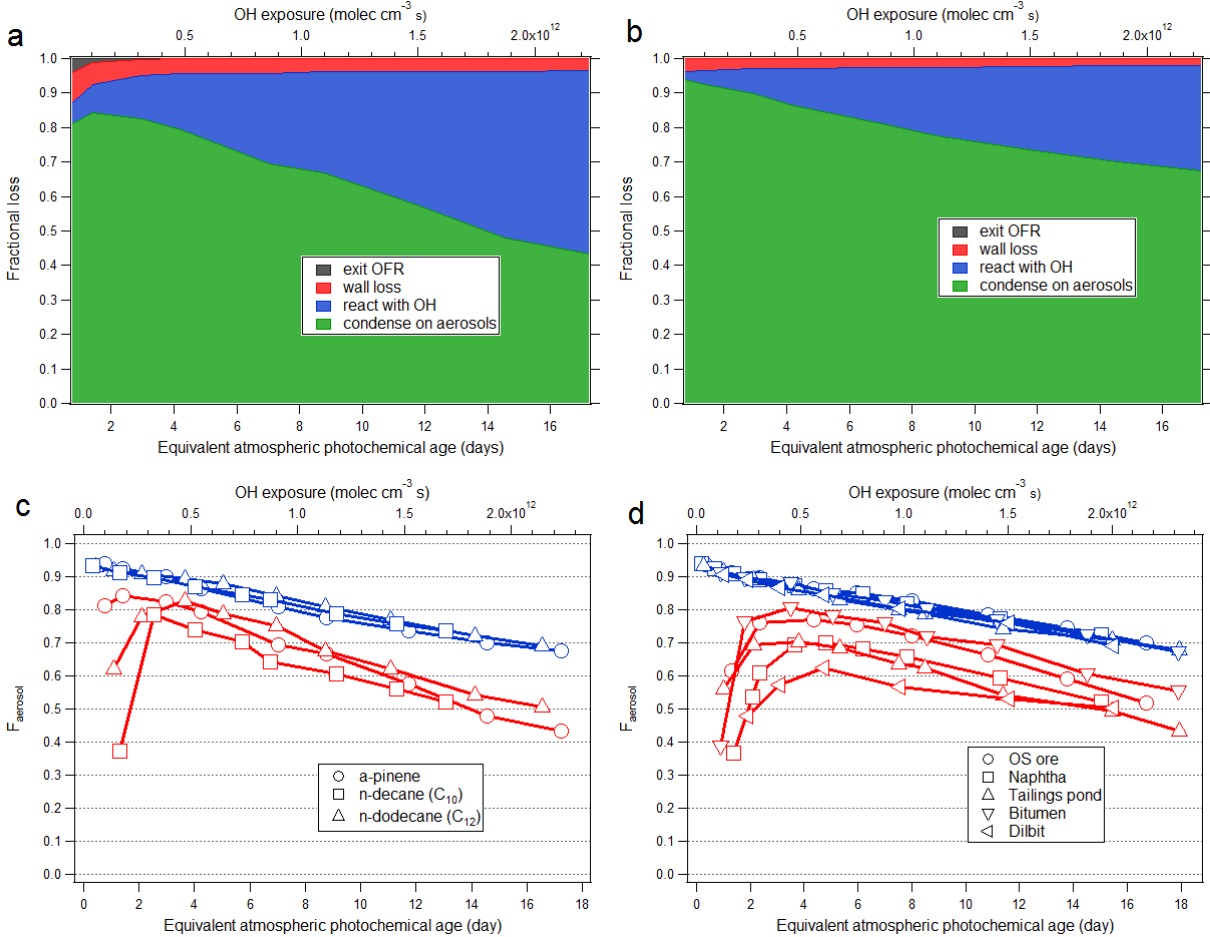

**Figure 3. (a, b): The modeled fate of LVOCs in the current OFR as a function of photochemical age, for α-pinene oxidation, in the absence (a) and presence (b) of AS seed particles. (c, d): Fraction of LVOCs that condense on aerosol (F$_{aerosol}$) in the OFR during the oxidation of the single precursors (c) and various OS-related precursors (d) (blue: seeded experiments; red: unseeded experiments).**




**Figure 4. (a):** SOA yields of OS-related precursors (OS ore, naphtha, tailings pond water, bitumen and dilbit) for unseeded experiments as a function of equivalent photochemical age and OH exposure. SOA yield of $C_7$, $C_{10}$ and $C_{12}$ *n*-alkanes, cyclodecane and decalin are also shown for comparison. **(b):** SOA yields as in (a) in the presence of ammonium sulfate seed particles. **(c):** SOA carbon and oxygen yields ($Y_C$ and $Y_O$) for the OS precursors of lowest and highest volatility (OS ore and naphtha solvent) compared to diesel and crude oil (Lambe et al., 2012). **(d):** Yield enhancement factor due to seed particles for OS-related precursors; dashed lines are exponential fittings for naphtha and OS ore data. Error bars indicate ±1σ uncertainty in measurements.





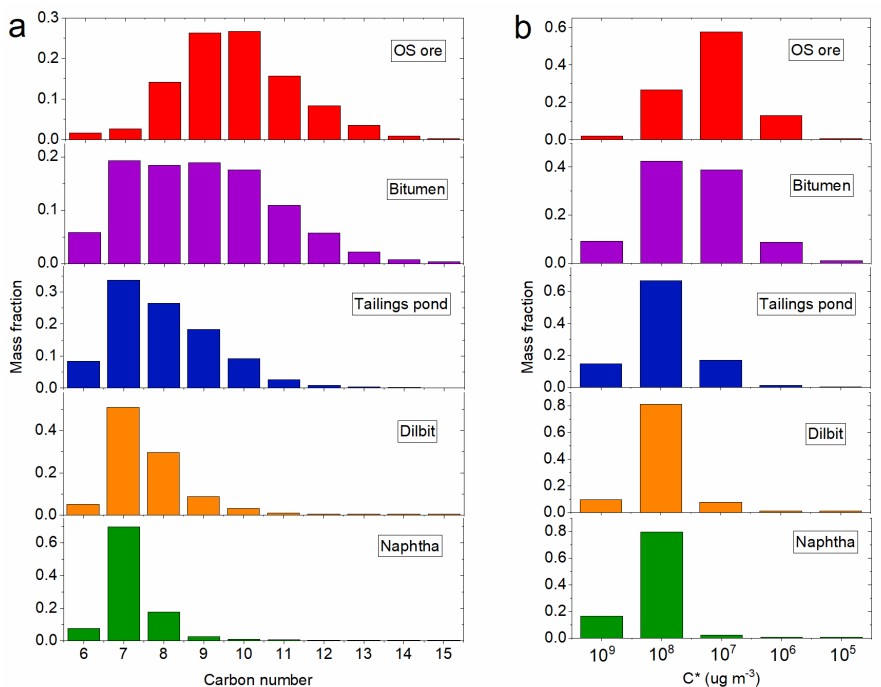

**Figure 5. Volatility distribution of the OS-related precursors binned by carbon number (a) and effective saturation concentration C\* (b).**





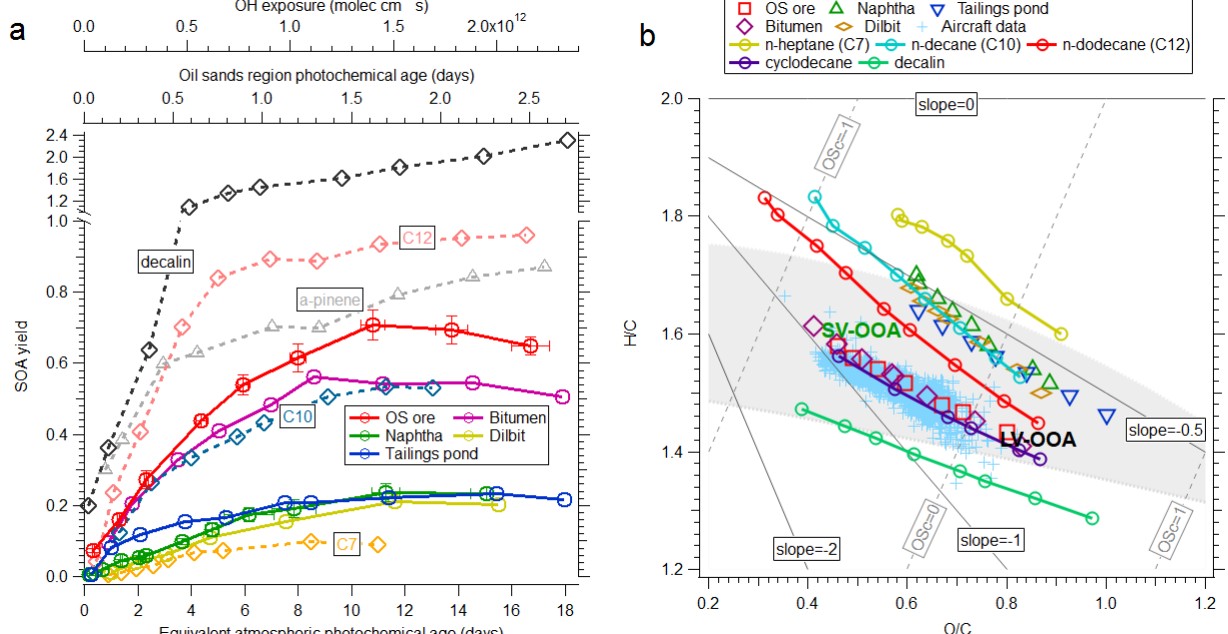

**Figure 6. (a): LVOCs fate corrected SOA yield of OS-related precursors, alkanes and α-pinene for AS seeded experiments. Representative error bars indicate ±1σ uncertainty in measurements. (b): Van Krevelen diagram for the SOA formed from OS-related precursors, selected alkanes and recent aircraft data in oil sands plumes (Liggio et al., 2016). The shaded area represents the elemental ratio space associated with ambient OOA (Ng et al., 2011).**

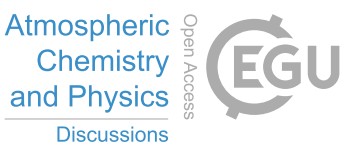

**Table 1. Initial concentrations, maximum SOA mass concentrations and maximum yields of OS-related precursors and selected compounds.**

| Precursor name | Elemental formula | Initial carbon concentration (ppbC) | | Maximum M_SOA (μg m⁻³) | | Maximum uncorrected yield[a] | | Maximum corrected yield[a] | |
| --- | --- | --- | --- | --- | --- | --- | --- | --- | --- |
| | | unseeded | seeded | unseeded | seeded | unseeded | seeded | unseeded | seeded |
| OS ore | - | 137 | 137 | 35.4 | 46.3 | 0.444 (14) | 0.581 (14) | 0.644 (17.8) | 0.708 (14) |
| Naphtha | - | 674 | 677 | 46.7 | 75.1 | 0.118 (14.6) | 0.191 (14.6) | 0.193 (19.5)[b] | 0.235 (14.6) |
| Tailings pond | - | 291 | 288 | 22.7 | 29.8 | 0.134 (9.72) | 0.176 (20) | 0.216 (20) | 0.279 (23.3)[b] |
| Bitumen | - | 218 | 201 | 44.8 | 54.7 | 0.353 (14.5) | 0.470 (11.1) | 0.494 (23.2)[b] | 0.545 (11.1) |
| Dilbit | - | 710 | 710 | 41.0 | 69.2 | 0.099 (20.1)[b] | 0.167 (15.1) | 0.177 (20.1)[b] | 0.209 (15.1) |
| n-Heptane | $C_7H_{16}$ | 1675 | 1671 | 43.1 | 85.2 | 0.044 (11) | 0.087 (11) | 0.076 (11) | 0.099 (11) |
| n-Decane | $C_{10}H_{22}$ | 211.6 | 233.9 | 36.3 | 57.8 | 0.295 (11.8) | 0.426 (14.6) | 0.497 (16.9)[b] | 0.534 (14.6) |
| n-Dodecane | $C_{12}H_{26}$ | 109.8 | 114.7 | 42.2 | 51.7 | 0.663 (9) | 0.778 (9) | 0.906 (21.4)[b] | 0.960 (21.4)[b] |
| Cyclodecane | $C_{10}H_{20}$ | 60.3 | - | 56.6 | - | 1.639 (19) | - | 2.121 (23)[b] | - |
| Decalin | $C_{10}H_{18}$ | 50.5 | 50.5 | 43.7 | 55.7 | 1.532 (23.4)[b] | 1.956 (23.4)[b] | 2.004 (23.4)[b] | 2.298 (23.4)[b] |
| α-Pinene | $C_{10}H_{16}$ | 137 | 137 | 38.7 | 48.0 | 0.508 (11.4) | 0.630 (22.3)[b] | 0.731 (22.3)[b] | 0.872 (22.3)[b] |

a. The number shown in the brackets is the corresponding OH exposure ($10^{11}$ molecules cm⁻³ s).

b. The SOA yield does not reach a maximum over OH exposure range, as such the highest OH exposure is shown here.



**Table 2. Comparison of experimental conditions and SOA yields with previous studies.**

| Precursor name | Mseed (μg m⁻³) | [precursor] (ppb) | Msoa (μg m⁻³) | SOA yield | OHexp (10¹¹ molec cm⁻³ s) | Reactor[c] | Reference |
|---|---|---|---|---|---|---|---|
| α-pinene | 0 | 41-100 | - | 0.35[a] | 5.57 | OFR | Lambe et al. (2015) |
| α-pinene | 0 | 50.6 | 90 | 0.32[a] | 6.6 | OFR | Chen et al. (2013) |
| α-pinene | 0 | 13.7 | 37.9 | 0.50[b] | 5.44 | OFR | This study |
| α-pinene | 13-19 | 44.5-47.7 | 63.5-76.6 | 0.26-0.29 | 0.91 | SC | Eddingsaas et al. (2012) |
| α-pinene | 14-21 | 13.8-47.5 | 29.3-121.3 | 0.38-0.46 | 1.21 | SC | Ng et al. (2007) |
| α-pinene | 12.6 | 19.6 | 34.1 | 0.35 | 0.52 | SC | Han et al. (2016) |
| α-pinene | 21 | 13.7 | 21.8 | 0.29[b] | 1 | OFR | This study |
| α-pinene | 10-60 | 192-200 | 540-570 | 0.55-0.56 | 2.6-3.6 | SC | Bruns et al. (2015) |
| α-pinene | 10-60 | 137-347 | 200-1000 | 0.31-0.67 | 3.5-11.9 | OFR | Bruns et al. (2015) |
| α-pinene | 21 | 13.7 | 41.9 | 0.55[b] | 3.9 | OFR | This study |
| n-decane | 0 | 102 | 231 | 0.39[a] | 5.3 | OFR | Lambe et al. (2012) |
| n-decane | 0 | 23.4 | 30.4 | 0.25[b] | 5.2 | OFR | This study |
| n-dodecane | 17-24 | 8.2-34 | 1.8-65 | 0.03-0.28 | 3.24 | SC | Loza et al. (2014) |
| n-dodecane | 21 | 9.6 | 24.3 | 0.37[b] | 2.72 | OFR | This study |

a. Maximum SOA yield.

b. The SOA yield at the OH exposure similar to above studies.

c. OFR: oxidation flow reactor; SC: smog chamber.