# Peer review of "Secondary organic aerosol formation from $\alpha$ -pinene, alkanes and oil sands related precursors in a new oxidation flow reactor"

_Atmospheric Chemistry and Physics, 2019_

## Referee Comment (RC1) · Anonymous Referee #1 · 24 May 2019

Li et al. present the evaluation of a custom ECCC-OFR design by performing characterization studies that included measurements of size-dependent particle transmission efficiency and yields of SOA generated from OH oxidation of $\alpha$-pinene and C7, C10, and C12 n-alkanes in the presence and absence of ammonium sulfate seeds. Results are compared to those obtained with other OFRs and environmental chambers. Unlike in previous OFR studies, alkane-generated SOA did not exhibit a decrease in yield at high OH exposure due to fragmentation reactions. The ECCC-OFR is then used to investigate the SOA formation potential following OH oxidation of materials obtained from oil sands operations in Alberta. Cyclic alkanes are implicated as the most important class of precursors in the oil sands samples. Overall, the manuscript reads well. Given the emergence of OFRs as a technique to characterize SOA formation, and the application of the ECCC-OFR to study the aging of environmentally-relevant VOC mixtures that are emitted during oil sands extraction activities, I would support eventual publication of this manuscript in ACP. However, in its current form, I have reservations about assumptions that are made regarding laminar flow behavior, reduced wall losses compared to other OFRs, and SOA yield calculations that are heavily reliant on offline measurements of SOA precursor concentrations. In my opinion these assumptions are not adequately justified based on the current information that is given, and any related conclusions made about ECCC-OFR performance compared to other OFRs are uncertain at present.

**General Comments**

1. Recent OFR applications and modeling studies have demonstrated the utility of 185 nm radiation in OFRs due to ease of use in the field and due to additional $HO_x$ generation via $H_2O + h\nu(185) \rightarrow$ H + OH, H + $O_2 \rightarrow$ $HO_2$. Here, the authors specifically mention that they chose to use mercury lamps that exclude the 185 nm emission line. Please explain the reasons for this choice.

2. The actinic flux at 254 nm is an important OFR characteristic that, unless I missed it, was never measured or calculated. It would be worthwhile to calculate this value and compare to the other OFR designs that are mentioned. For example, a possibility that is never discussed is whether potential SOA photolysis at 254 nm (which is more potentially important at high UV intensity and OH exposure) might be less important in the ECCC-OFR than in the PAM OFR due to lower actinic flux. I am not necessarily convinced that this is the case, but it should be briefly discussed and ruled out if not applicable. The preferable method to quantify the actinic flux would be to photolyze a compound with known absorption cross section at 254 nm as a function of lamp voltage. At the least, I think the maximum actinic flux inside the ECCC-OFR could be estimated from the wattage of the UV lamps at full output normalized by the internal surface area, with the caveat that I am not to what extent the mirrored enclosure referred to on P4, L14-15 would influence this calculation.

3. In the ECCC-OFR, the authors state that an inlet with a cone angle of 30° is used to "minimize the establishment of jetting and recirculation in the OFR", which is steeper than the 15° cone angle used by Huang et al. (2017) and the 14° cone angle used by Ihalainen et al. (2019). Whereas both of those studies employed CFD simulations to optimize their OFR design, there are no corresponding simulations of the ECCC-OFR fluid dynamics that support the 30° cone angle used here. Please provide supporting calculations and/or residence time distribution measurements supporting the claim that laminar flow is achieved and jetting/recirculation is not present when using a 30° cone angle.

4. The authors hypothesize that wall interactions are minor in the ECCC-OFR based on a calculation of the diffusion timescale (1400 sec) that is much longer than the residence time (120 sec) (P7, L1-8). Applying the same calculation to the PAM OFR, which has an inner radius of 10.2 cm, yields a diffusion timescale of 1474 sec. Given similar residence times and diffusion timescales, this line of reasoning would suggest similar wall interactions between the two systems. However, large-scale dispersion and recirculation inside OFRs (e.g. Lambe et al., 2011; Huang et al., 2017) complicates this sort of simple diffusion-based calculation. Later on (P9, L6-7), the authors speculate that higher SOA yields and less fragmentation are observed in the ECCC-OFR because of reduced wall interactions compared to other OFRs. This might be the case, but it is not supported by the logic presented above. This conclusion should be supported with a corresponding residence time distribution measurement and comparison to the RTD expected for ideal laminar flow, which was not performed here (P7, L9). In my opinion this is a critical oversight that should be addressed. Additionally, I suggest measuring the yield of sulfuric acid generated from OH oxidation of $SO_2$ and comparing the result to other OFRs. Because sulfuric acid is not affected by photolysis or fragmentation, any difference in sulfuric acid yields between OFRs should be directly related to wall losses/interactions.

**Specific/Technical Comments**

5. **P4, L16**: Please specify the $O_3$ mixing ratio (or range of $O_3$ mixing ratios) that was added to the OFR in these studies.

6. **P5, L5-7**: Because precursor concentrations were only obtained in offline measurements, how did the authors determine that the precursor concentrations remained constant and precise during the OFR experiments? As written, in the absence of other supporting/independent measurements this seems to be a major assumption and potential source of uncertainty in the SOA yield calculations.

7. **Figure 1 and Section 3.1.1.**: The Lambe et al. 2011 reference used a Pyrex chamber, where wall losses of charged particles are higher than chambers made of conductive materials due to charge buildup on nonconductive surfaces. A better reference/comparison here would be to use the data from Figure S1 of Karjaranen et al. (2016) which used an aluminum chamber with conductive coating. Their particle transmission data is shown below for reference. Please modify the discussion and figure accordingly.

[Figure]

Figure S1. Primary particle losses in a similar PAM chamber that was used in the study.

8. **P15, L7**: The authors state: "all future OFR experiments should be conducted with seed particles to obtain more relevant qualitative and quantitative data." I suggest making this statement in the specific context of laboratory SOA yield studies, as not all OFR experiments are intended to measure SOA yields and because addition of seed particles in ambient OFR experiments is not necessarily always desirable or practical.

9. **P11, L30 and Figure 4c**: Lambe et al. (2012) do not report absolute SOA yields from OH oxidation of diesel fuel and crude oil so it is unclear where this statement originates from.

10. **Figure 1**: It may be worth adding particle transmission data from Ihalainen et al. (2019) to this figure. Also, how much passivation time is required to obtain 100% transmission efficiency of C7, C10 and C12 alkanes, and at what mixing ratios are they introduced to the ECCC-OFR?

11. **Figure 3**: I think this could be moved to the Supplement.

12. **Figures 4-6 and related text**: I suggest a reorganization to improve clarity and flow. First, move the current Figure 5 to the Supplement or to Methods. Second, combine the current Figure 6a with the current Figures 4a and 4b into a single 3-panel figure. Third, move the current Figure 4d into a separate figure and place between current Figures 4 and 6.

**References**

M. Ihalainen, P. Tiitta, H. Czech, P. Yli-Pirilä, A. Hartikainen, M. Kortelainen, J. Tissari, B. Stengel, M. Sklorz, H. Suhonen, H. Lamberg, A. Leskinen, A. Kiendler-Scharr, H. Harndorf, R. Zimmermann, J. Jokiniemi, and O. Sippula (2019) A novel high-volume Photochemical Emission Aging flow tube Reactor (PEAR), Aerosol Science and Technology, 53:3, 276-294, DOI: 10.1080/02786826.2018.1559918.

Karjalainen, P., Timonen, H., Saukko, E., Kuuluvainen, H., Saarikoski, S., Aakko-Saksa, P., Murtonen, T., Bloss, M., Dal Maso, M., Simonen, P., Ahlberg, E., Svenningsson, B., Brune, W. H., Hillamo, R., Keskinen, J., and Rönkkö, T.: Time-resolved characterization of primary particle emissions and secondary particle formation from a modern gasoline passenger car, Atmos. Chem. Phys., 16, 8559-8570, https://doi.org/10.5194/acp-16-8559-2016, 2016.

---

## Referee Comment (RC2) · Anonymous Referee #2 · 27 May 2019

**Summary:**

The oil sands (OS) in Alberta, Canada provide a significant source of SOA, necessitating lab studies to isolate contributions from different sources and chemical reactions. To address this knowledge gap, the authors use a custom oxidative flow reactor (OFR) to mimic different degrees of atmospheric oxidative aging for emissions from different OS-related precursors. In this work, the authors introduce the ECCC-OFR through single-species precursor experiments to assess the impacts of gas and particle wall losses and seeding, then use the ECCC-OFR to evaluate differences in OS-related SOA formation between several relevant sources. This is generally a clearly written manuscript, with compelling results that contribute important knowledge for both OS SOA chemistry as well as future OFR laboratory studies.

**General Comments**

[1] In the introduction (page 2, lines 22-24), the authors state that organic gases from the OS are mainly alkanes that react with the OH radical. However, one of the precursors that the authors use and discuss in the introduction is α-pinene. The choice of α-pinene is confusing in this context without further justification. From the manuscript, it seems that α-pinene was chosen because it was convenient to compare OFR operation to other studies. Does α-pinene have additional relevance for SOA in the OS region? Either way, it would be helpful for the author to address this choice early on in the manuscript. Additionally, under the ECCC-OFR operating conditions for these experiments (i.e., precursor concentrations, ozone concentrations), is there potential for the interfering α-pinene + ozone reaction to contribute significantly to SOA yields?

[2] Wall losses (Section 3.1.1): The authors state that vapor wall losses are likely minimal based on the diffusion timescale relative to the residence time within the reactor, then state the critical assumption that flow in the reactor is ideally laminar. Is this assumption solely based on fluid dynamics information from previously designed OFRs? The authors cite CFD done by Huang et al. (2017) for the CPOT on page 4 (lines 6-7) to justify the assumption, but I'm curious as to how the differences between the ECCC-OFR and the CPOT would change the fluid dynamics. For example, the ECCC-OFR has a straight outlet rather than a conical one like the CPOT. Is there potential for jetting or dead volume around the outlet? What are the benefits to sampling from the center line?

**Technical Comments**

[1] Page 2, Lines 19-20: The authors state that a single species approach to studying SOA formation is "impractical." To me, "impractical" implies some sort of logistical difficulty and sells the point short. I'd consider reframing this sentence to emphasize atmospheric relevance for the OS, which is critical to consider when performing lab studies.

[2] Page 2, Lines 21-22: Consider restructuring this sentence for clarity. Perhaps "Precursor emissions occur throughout the OS surface mining and processing production cycle, and they originate from sources including…"

[3] Page 2, Line 24: Define "OH" as "hydroxyl radicals (OH)" before using the abbreviation.

[4] Page 2, Line 28: "Complimentary" should be "complementary." This spelling should also be changed on page 3, line 13.

[5] Page 3, Line 6: Replace the semicolon after "vary" with a comma.

[6] Page 4, Line 11: Replace "Hg" with "mercury."

[7] Page 5, Line 5: Define the THC acronym here.

[8] Figure 1: Consider matching the color of the top and right axes to the alkane data points to visually distinguish the gas-phase data from the particle-phase data.

[9] Page 8, line 9: The sentence starting with "This despite" is not a full sentence.

[10] Page 10, line 21: Replace the semicolon after "mixtures" with a comma.

[11] Page 10, line 25: It would be helpful to cite the specific section in supporting information so the reader can easily flip to it as needed.

[12] Figure 4a and 4b: Consider emphasizing the different y axis scales between the two panels in either the text or the figure caption. Otherwise, the differences between seeded and non-seeded results can be difficult to pick out visually.

[13] I would be interested to see the AMS mass spectra for each OS-related oxidation experiment, perhaps in the supplement.

---

## Author Comment (AC1) · 27 Jun 2019

**Response to the comments of Anonymous Referee #1**

Li et al. present the evaluation of a custom ECCC-OFR design by performing characterization studies that included measurements of size-dependent particle transmission efficiency and yields of SOA generated from OH oxidation of α-pinene and C7, C10, and C12 n-alkanes in the presence and absence of ammonium sulfate seeds. Results are compared to those obtained with other OFRs and environmental chambers. Unlike in previous OFR studies, alkane-generated SOA did not exhibit a decrease in yield at high OH exposure due to fragmentation reactions. The ECCC-OFR is then used to investigate the SOA formation potential following OH oxidation of materials obtained from oil sands operations in Alberta. Cyclic alkanes are implicated as the most important class of precursors in the oil sands samples. Overall, the manuscript reads well. Given the emergence of OFRs as a technique to characterize SOA formation, and the application of the ECCC-OFR to study the aging of environmentally-relevant VOC mixtures that are emitted during oil sands extraction activities, I would support eventual publication of this manuscript in ACP. However, in its current form, I have reservations about assumptions that are made regarding laminar flow behavior, reduced wall losses compared to other OFRs, and SOA yield calculations that are heavily reliant on offline measurements of SOA precursor concentrations. In my opinion these assumptions are not adequately justified based on the current information that is given, and any related conclusions made about ECCC-OFR performance compared to other OFRs are uncertain at present.

Response: We thank Anonymous Referee #1 for the review and the positive evaluation of our manuscript. We have fully considered the comments, responded to these comments below in blue text and made the associated revisions to the manuscript as shown in red text. The response and changes are listed below.

**General Comments**

1. Recent OFR applications and modeling studies have demonstrated the utility of 185 nm radiation in OFRs due to ease of use in the field and due to additional $HO_x$ generation via $H_2O + h\nu(185) \rightarrow H + OH$, $H + O_2 \rightarrow HO_2$. Here, the authors specifically mention that they chose to use mercury lamps that exclude the 185 nm emission line. Please explain the reasons for this choice.

Response: The lamps are located outside of the fused quartz reactor, and the transmittance of 185 nm UV light through fused quartz is very small, which limits the application of 185 nm UV lamps in our OFR. We calculate that only ~5% of the 185 nm light can be transmitted through the fused quartz wall; hence, 185 nm lamps were not used. Placing the lamps on the inside of the OFR to avoid this problem would have introduced additional surface areas and turbulence, thus negating some of the other advantages of this OFR, such as reduced wall loses. In addition, interior 185 nm lights would have made temperature control difficult. Exterior lamps allow the temperature of the OFR to be maintained accurately.

To address these points we added the following text in the revised manuscript (P4, L18-23):

"Recent OFR applications and modeling studies have demonstrated the utility of 185 nm radiation in OFRs due to ease of use in the field and due to additional OH and $HO_2$ generation (Li et al., 2015;Palm et al., 2016). However, the fused quartz tubes of ECCC-OFR limit the application of such lamps due to the low transmittance of 185 nm radiation (~5%), and placement of lamps on the interior of the OFR are likely to increase turbulence and wall losses within the OFR, and limit overall OFR temperature control. Consequently, 254 nm radiation lamps were used."

2. The actinic flux at 254 nm is an important OFR characteristic that, unless I missed it, was never measured or calculated. It would be worthwhile to calculate this value and compare to the other OFR designs that are mentioned. For example, a possibility that is never discussed is whether potential SOA photolysis at 254 nm (which is more potentially important at high UV intensity and OH exposure) might be less important in the ECCC-OFR than in the PAM OFR due to lower actinic flux. I am not necessarily convinced that this is the case, but it should be briefly discussed and ruled out if not applicable. The preferable method to quantify the actinic flux would be to photolyze a compound with known absorption cross section at 254 nm as a function of lamp voltage. At the least, I think the maximum actinic flux inside the ECCC-OFR could be estimated from the wattage of the UV lamps at full output normalized by the internal surface area, with the caveat that I am not to what extent the mirrored enclosure referred to on P4, L14-15 would influence this calculation.

Response: We have now determined the maximum photon flux (with 4 lamps on) based upon the measured ozone decay and OH exposure (without precursor injection) combined with a photochemical box model characterizing radical chemistry in OFRs (Oxidation Flow Reactor Exposure Estimator 3.1) (Li et al., 2015;Peng et al., 2018). The input photon flux of the model was adjusted to match the measured ozone decay and OH exposure, which resulted in a maximum photon flux estimate of ~$1.9 \times 10^{16}$ photons cm$^{-2}$ s$^{-1}$. This photon flux is similar to the PEAR OFR ($2.3 \times 10^{16}$ photons cm$^{-2}$ s$^{-1}$) using the same estimation method (Ihalainen et al., 2019), and about three times that reported for the PAM ($6.4 \times 10^{15}$ photons cm$^{-2}$ s$^{-1}$) (Lambe et al., 2017). Consequently, we can rule out the possibility of lower SOA photolysis in the ECCC-OFR as the reason for the higher yields in the ECCC-OFR compared to the PAM OFR.

The above content has been added to the revised manuscript (P4, L23-29).

3. In the ECCC-OFR, the authors state that an inlet with a cone angle of 30° is used to "minimize the establishment of jetting and recirculation in the OFR", which is steeper than the 15° cone angle used by Huang et al. (2017) and the 14° cone angle used by Ihalainen et al. (2019). Whereas both of those studies employed CFD simulations to optimize their OFR design, there are no corresponding simulations of the ECCC-OFR fluid dynamics that support the 30° cone angle used here. Please provide supporting calculations and/or residence time distribution measurements supporting the claim that laminar flow is achieved and jetting/recirculation is not present when using a 30° cone angle.

Response: The reviewer is not comparing the same angles in the above comment. The cone angle mentioned in our paper (i.e., 30°) and Huang et al. (2017) is the full cone angle, while Ihalainen et al. (2019) reported the half cone angle. If the same angles are compared, then the half cone angles for the various OFRs are: 7.5° for Huang et al. (2017), 14° for Ihalainen et al. (2019), and 15° for the ECCC-OFR of this study. Hence, the cone angle of ECCC-OFR is very similar to that of the PEAR OFR (Ihalainen et al., 2019). This cone angle comparison was added to Sect. S1 of the Supplement.

To assess the near laminar flow of the ECCC-OFR, computational fluid dynamics (CFD) simulations were performed using ANSYS Fluent software (Version 2019 R2) in three dimensions to characterize the flow field inside the ECCC-OFR. Hybrid tetrahedral–hexahedral mesh consisting of $5.7 \times 10^5$ computation cells were used. Turbulence was modeled using a realizable k-epsilon model. The simulation results are shown in Fig. S4. It is shown in Fig. S4a that the flow velocity distribution in the reactor is generally uniform. A high velocity is observed only near the inlet, but reduces to the average velocity in

the conical diffuser. The velocity distribution here indicates that jetting is much weaker in ECCC-OFR compared to PAM (Mitroo et al., 2018). Fig. S4b indicates that the flow field is quite good in ECCC-OFR, with a small recirculation zone, similar to previous studies using a conical diffusion inlet (Huang et al., 2017;Ihalainen et al., 2019), but much better than PAM (Mitroo et al., 2018). These CFD simulations were added to Sect. S2 of the Supplement.

[Figure]

Figure S4. CFD simulation results: (a) velocity distribution; (b) vectors showing flow field. The red lines in (b) indicate the areas with recirculation.

4. The authors hypothesize that wall interactions are minor in the ECCC-OFR based on a calculation of the diffusion timescale (1400 sec) that is much longer than the residence time (120 sec) (P7, L1-8). Applying the same calculation to the PAM OFR, which has an inner radius of 10.2 cm, yields a diffusion timescale of 1474 sec. Given similar residence times and diffusion timescales, this line of reasoning would suggest similar wall interactions between the two systems. However, large-scale dispersion and recirculation inside OFRs (e.g. Lambe et al., 2011; Huang et al., 2017) complicates this sort of simple diffusion-based calculation. Later on (P9, L6-7), the authors speculate that higher SOA yields and less fragmentation are observed in the ECCC-OFR because of reduced wall interactions compared to other OFRs. This might be the case, but it is not supported by the logic presented above. This conclusion should be supported with a corresponding residence time distribution measurement and comparison to the RTD expected for ideal laminar flow, which was not performed here (P7, L9). In my opinion this is a critical

oversight that should be addressed. Additionally, I suggest measuring the yield of sulfuric acid generated from OH oxidation of $SO_2$ and comparing the result to other OFRs. Because sulfuric acid is not affected by photolysis or fragmentation, any difference in sulfuric acid yields between OFRs should be directly related to wall losses/interactions.

Response: Based upon the CFD simulation results above, we know that the flow field in ECCC-OFR is not perfectly ideal laminar flow, though it is significantly better than previous OFRs with a straight inlet, e.g., PAM (Mitroo et al., 2018). Hence, our assumption based on ideal laminar flow (using a diffusion timescale compared to the residence time to infer the gas-wall interactions) was removed in our revised paper (P7, L16-26).

The residence time distribution (RTD) was measured for ECCC-OFR and compared to ideal laminar flow in Fig. S5. The RTD was characterized by injecting a constant flow rate of $CO_2$ (10 s) into the OFR. The $CO_2$ concentration was then monitored from the sampling outlet of OFR with a $CO_2$ analyzer (Li-Cor LI-840A). The RTD was calculated from the differential $CO_2$ as a function of time elapsed since the start of injection. Fig. S5 indicates that the residence time associated with the $CO_2$ maximum intensity for the measured RTD and the ideal laminar flow RTD are in good agreement, and improved over the PAM and TPOT (Lambe et al., 2011). The shape of measured RTD before ~100 s is similar to the ideal laminar flow RTD, but slightly wider, which is likely due to dispersion. After ~100 s, the measured decrease in $CO_2$ is slower than for ideal laminar flow, which is likely due to recirculation in the OFR (Fig. S4).

This paragraph and associated figure were added to Sect. S2 of the Supplement.

[Figure]

Figure S5. Residence time distribution (RTD) of $CO_2$ in ECCC-OFR compared to ideal laminar flow.

As suggested by the reviewer, the OH oxidation of $SO_2$ was performed in the ECCC-OFR. The $SO_2$ concentrations used were in the range of 24-63 ppb and the OH exposure was in the range of 3-10 $\times 10^{11}$ molec $cm^{-3}$ s, which are similar to those used in a previous PAM study (Lambe et al., 2011). The yield of sulfuric acid ($Y_{H_2SO_4}$) was calculated using the mass fraction of $H_2SO_4$ in particles ($x_{H_2SO_4}$), the SMPS-measured particle volume ($V_{H_2SO_4 \cdot H_2O}$, $nm^3$ $cm^{-3}$), the density of the particles ($\rho_{H_2SO_4 \cdot H_2O}$, g $cm^{-3}$), and the measurement of the reacted $SO_2$ ($\Delta SO_2$, ppb), using an approach which has been described in detail previously (Lambe et al., 2011):

$$Y_{H_2SO_4} = \frac{x_{H_2SO_4} \times V_{H_2SO_4 \cdot H_2O} \times \rho_{H_2SO_4 \cdot H_2O}}{3.95 \times \Delta SO_2}$$

The $x_{H_2SO_4}$ and $\rho_{H_2SO_4 \cdot H_2O}$ were estimated using the Extended Aerosol Inorganic Thermodynamic Model (E-AIM) I (http://www.aim.env.uea.ac.uk/aim/aim.php) (Carslaw et al., 1995). The dry yield of $H_2SO_4$ from OH oxidation of $SO_2$ is 3.95 µg m$^{-3}$ per ppb $SO_2$ reacted (Lambe et al., 2011); hence $Y_{H_2SO_4}$ is expected to be 100% without wall losses. As shown in Fig. S6, the yield of sulfuric acid is 100±4% in this study, which is in agreement with the expected yield. The yield here is significantly higher than that obtained in previous OFR study using similar $SO_2$ concentrations and OH exposures (PAM and TPOT), which are mainly in the range of ~15%-50% (Lambe et al., 2011). This may be a result of lower wall losses in current OFR for gas-phase sulfuric acid and/or particles. Given that sulfuric acid is not impacted by photolysis or fragmentation, the result here suggest that wall losses/interactions within the ECCC-OFR are significantly lower than previous OFRs that utilize straight inlets (PAM and TPOT).

The above paragraph was added to the revised manuscript (P7, L27-33; P8, L1-3) and the Supplement (Sect. S5).

[Figure]

Figure S6. $H_2SO_4$ yields as a function of particle diameter in the ECCC-OFR and previous OFRs (Lambe et al., 2011).

**Specific/Technical Comments**

5. **P4, L16**: Please specify the $O_3$ mixing ratio (or range of $O_3$ mixing ratios) that was added to the OFR in these studies.

Response: The $O_3$ mixing ratio was ~12 ppm (now noted on P4, L17).

6. **P5, L5-7**: Because precursor concentrations were only obtained in offline measurements, how did the authors determine that the precursor concentrations remained constant and precise during the OFR experiments? As written, in the absence of other supporting/independent measurements this seems to be a major assumption and potential source of uncertainty in the SOA yield calculations.

Response: The THC concentration was measured before and after each experiment, and the difference was found to be less than 5%, representing a small uncertainty in the experiments. In addition, the precursor concentration was qualitatively checked during each experiment by repeating the same UV light intensity several times. For example, the UV lamp voltages were stepped from 120→50→60→120→70→80→120 etc... The absolute amount of SOA formed at the repeated light intensities varied by less than 5%, indicating that the THC precursor concentration was rather stable during experiments.

We have added the following text in the revised manuscript (P5, L22-24): "The THC concentration was measured before and after each experiment, resulting in differences of less than 5%. In addition, the magnitude of SOA formed for repeated experiments at the same light intensity varied by less than 5%, further indicating the stability of the precursor concentration over time."

7. **Figure 1 and Section 3.1.1.**: The Lambe et al. 2011 reference used a Pyrex chamber, where wall losses of charged particles are higher than chambers made of conductive materials due to charge buildup on nonconductive surfaces. A better reference/comparison here would be to use the data from Figure S1 of Karjaranen et al. (2016) which used an aluminum chamber with conductive coating. Their particle transmission data is shown below for reference. Please modify the discussion and figure accordingly.

[Figure]

Figure S1. Primary particle losses in a similar PAM chamber that was used in the study.

Response: Particle transmission efficiencies from Karjalainen et al. (2016) and Ihalainen et al. (2019) were added in Figure 1 (now Figure 2).

The corresponding discussion was changed in the revised manuscript (P6, L28-31; P7, L1-2):

"The $P_{trans}$ of other OFRs are also shown in Fig. 2 for comparison and indicates that the current $P_{trans}$ is similar to that of the TSAR (TUT Secondary Aerosol Reactor) (Simonen et al., 2017) and PEAR (Ihalainen et al., 2019), likely due to the similarity in design (i.e., cone shaped inlet and sampling from the center-line, see Sect. S1 in Supplement). Conversely, the $P_{trans}$ of the TPOT (Toronto Photo-Oxidation Tube), PAM-glass (PAM reactor with glass wall) (Lambe et al., 2011), PAM-metal (PAM reactor with metal wall) (Karjalainen et al., 2016), and CPOT (Caltech Photooxidation Flow Tube) (Huang et al., 2017) are 15-85%, 20-60%, 10-25%, and 20-45% lower respectively than the ECCC-OFR across a range of particle sizes."

[Figure]

Figure 2. Particle (left and bottom axis) and gas (right and top axis) transmission efficiencies ($P_{trans\ and}\ G_{trans}$) for the ECCC-OFR. Particle transmission efficiencies of other OFRs are shown for comparison: PAM-glass and TPOT (Lambe et al., 2011), PAM-metal (Karjalainen et al., 2016), TSAR (Simonen et al., 2017), CPOT (Huang et al., 2017) and PEAR (Ihalainen et al., 2019).

8. **P15, L7**: The authors state: "all future OFR experiments should be conducted with seed particles to obtain more relevant qualitative and quantitative data." I suggest making this statement in the specific context of laboratory SOA yield studies, as not all OFR experiments are intended to measure SOA yields and because addition of seed particles in ambient OFR experiments is not necessarily always desirable or practical.

Response: The line in has been revised as: "This suggests that future laboratory OFR experiments studying SOA yields should be conducted with seed particles to obtain more relevant qualitative and quantitative data." (P15, L23-24).

9. **P11, L30 and Figure 4c**: Lambe et al. (2012) do not report absolute SOA yields from OH oxidation of diesel fuel and crude oil so it is unclear where this statement originates from.

Response: The carbon and oxygen yields of diesel fuel and crude oil in Figure 4c (now is Figure 5b) are what are referred to as the "normalized yields". We do not use the absolute yields here as we only compare the relative change of yields (relative to the maximum yield) among different precursors.

We changed the figure caption to reflect this: "…normalized $Y_C$ and $Y_O$ for diesel and crude oil…" (P24, L6).

10. **Figure 1**: It may be worth adding particle transmission data from Ihalainen et al. (2019) to this figure. Also, how much passivation time is required to obtain 100% transmission efficiency of C7, C10 and C12 alkanes, and at what mixing ratios are they introduced to the ECCC-OFR?

Response: Particle transmission data from Ihalainen et al. (2019) are now added to Figure 1 (now Figure 2), see above response to Comment 7 for details.

The passivation time was 5-10 min, and the mixing ratio was 300-500 ppb for these alkanes (now noted in P7, L10-11).

11. **Figure 3**: I think this could be moved to the Supplement.

Response: This figure provides important information about the LVOC fates in ECCC-OFR, and can help to better understand the final corrected SOA yields from OS precursors. Hence, we have decided to keep it in the main manuscript.

12. **Figures 4-6 and related text**: I suggest a reorganization to improve clarity and flow. First, move the current Figure 5 to the Supplement or to Methods. Second, combine the current Figure 6a with the current Figures 4a and 4b into a single 3-panel figure. Third, move the current Figure 4d into a separate figure and place between current Figures 4 and 6.

Response: Figure 5 is now Figure 1 and is described in Methods (P5, L27-28): "The chromatogram and the derived VD of the OS-related precursors are shown in Fig. S2 and Fig. 1 and discussed in detail in Sect. S4 of the Supplement."

Figures 1-3 are now Figures 2-4; Figure 6a is now moved into Figure 5 as Figure 5d; Figure 4d is now Figure 6; Figure 6b is now Figure 7.

The corresponding text is revised to match the Figure numbers above.

**References**

Carslaw, K. S., Clegg, S. L., and Brimblecombe, P.: A Thermodynamic Model of the System HCl-HNO3-H2SO4-H2O, Including Solubilities of HBr, from <200 to 328 K, The Journal of Physical Chemistry, 99, 11557-11574, 10.1021/j100029a039, 1995.

Huang, Y., Coggon, M. M., Zhao, R., Lignell, H., Bauer, M. U., Flagan, R. C., and Seinfeld, J. H.: The Caltech Photooxidation Flow Tube reactor: design, fluid dynamics and characterization, Atmospheric Measurement Techniques, 10, 839-867, 10.5194/amt-10-839-2017, 2017.

Ihalainen, M., Tiitta, P., Czech, H., Yli-Pirilä, P., Hartikainen, A., Kortelainen, M., Tissari, J., Stengel, B., Sklorz, M., Suhonen, H., Lamberg, H., Leskinen, A., Kiendler-Scharr, A., Harndorf, H., Zimmermann, R., Jokiniemi, J., and Sippula, O.: A novel high-volume Photochemical Emission Aging flow tube Reactor (PEAR), Aerosol Science and Technology, 53, 276-294, 10.1080/02786826.2018.1559918, 2019.

Karjalainen, P., Timonen, H., Saukko, E., Kuuluvainen, H., Saarikoski, S., Aakko-Saksa, P., Murtonen, T., Bloss, M., Dal Maso, M., Simonen, P., Ahlberg, E., Svenningsson, B., Brune, W. H., Hillamo, R., Keskinen, J., and Rönkkö, T.: Time-resolved characterization of primary particle emissions and secondary particle formation from a modern gasoline passenger car, Atmospheric Chemistry and Physics, 16, 8559-8570, 10.5194/acp-16-8559-2016, 2016.

Lambe, A., Massoli, P., Zhang, X., Canagaratna, M., Nowak, J., Daube, C., Yan, C., Nie, W., Onasch, T., Jayne, J., Kolb, C., Davidovits, P., Worsnop, D., and Brune, W.: Controlled nitric oxide production via O($^1$D) + N$_2$O reactions for use in oxidation

flow reactor studies, Atmospheric Measurement Techniques, 10, 2283-2298, 10.5194/amt-10-2283-2017, 2017.

Lambe, A. T., Ahern, A. T., Williams, L. R., Slowik, J. G., Wong, J. P. S., Abbatt, J. P. D., Brune, W. H., Ng, N. L., Wright, J. P., Croasdale, D. R., Worsnop, D. R., Davidovits, P., and Onasch, T. B.: Characterization of aerosol photooxidation flow reactors: heterogeneous oxidation, secondary organic aerosol formation and cloud condensation nuclei activity measurements, Atmospheric Measurement Techniques, 4, 445-461, 10.5194/amt-4-445-2011, 2011.

Li, R., Palm, B. B., Ortega, A. M., Hlywiak, J., Hu, W., Peng, Z., Day, D. A., Knote, C., Brune, W. H., de Gouw, J. A., and Jimenez, J. L.: Modeling the radical chemistry in an oxidation flow reactor: radical formation and recycling, sensitivities, and the OH exposure estimation equation, The journal of physical chemistry. A, 119, 4418-4432, 10.1021/jp509534k, 2015.

Mitroo, D., Sun, Y., Combest, D. P., Kumar, P., and Williams, B. J.: Assessing the degree of plug flow in oxidation flow reactors (OFRs): a study on a potential aerosol mass (PAM) reactor, Atmospheric Measurement Techniques, 11, 1741-1756, 10.5194/amt-11-1741-2018, 2018.

Palm, B. B., Campuzano-Jost, P., Ortega, A. M., Day, D. A., Kaser, L., Jud, W., Karl, T., Hansel, A., Hunter, J. F., Cross, E. S., Kroll, J. H., Peng, Z., Brune, W. H., and Jimenez, J. L.: In situ secondary organic aerosol formation from ambient pine forest air using an oxidation flow reactor, Atmospheric Chemistry and Physics, 16, 2943-2970, 10.5194/acp-16-2943-2016, 2016.

Peng, Z., Palm, B. B., Day, D. A., Talukdar, R. K., Hu, W., Lambe, A. T., Brune, W. H., and Jimenez, J. L.: Model Evaluation of New Techniques for Maintaining High-NO Conditions in Oxidation Flow Reactors for the Study of OH-Initiated Atmospheric Chemistry, ACS Earth and Space Chemistry, 2, 72-86, 10.1021/acsearthspacechem.7b00070, 2018.

Simonen, P., Saukko, E., Karjalainen, P., Timonen, H., Bloss, M., Aakko-Saksa, P., Rönkkö, T., Keskinen, J., and Dal Maso, M.: A new oxidation flow reactor for measuring secondary aerosol formation of rapidly changing emission sources, Atmospheric Measurement Techniques, 10, 1519-1537, 10.5194/amt-10-1519-2017, 2017.

---

## Author Comment (AC2) · 27 Jun 2019

**Response to the comments of Anonymous Referee #2**

**Summary:**

The oil sands (OS) in Alberta, Canada provide a significant source of SOA, necessitating lab studies to isolate contributions from different sources and chemical reactions. To address this knowledge gap, the authors use a custom oxidative flow reactor (OFR) to mimic different degrees of atmospheric oxidative aging for emissions from different OS-related precursors. In this work, the authors introduce the ECCC-OFR through single-species precursor experiments to assess the impacts of gas and particle wall losses and seeding, then use the ECCC-OFR to evaluate differences in OS-related SOA formation between several relevant sources. This is generally a clearly written manuscript, with compelling results that contribute important knowledge for both OS SOA chemistry as well as future OFR laboratory studies.

Response: We thank Anonymous Referee #2 for the review and the positive evaluation of our manuscript. We have fully considered the comments and made the associated revisions to our manuscript. The responses (blue text) and changes to the manuscript (red text) are listed below.

**General Comments**

[1] In the introduction (page 2, lines 22-24), the authors state that organic gases from the OS are mainly alkanes that react with the OH radical. However, one of the precursors that the authors use and discuss in the introduction is α-pinene. The choice of α-pinene is confusing in this context without further justification. From the manuscript, it seems that α-pinene was chosen because it was convenient to compare OFR operation to other studies. Does α-pinene have additional relevance for SOA in the OS region? Either way, it would be helpful for the author to address this choice early on in the manuscript. Additionally, under the ECCC-OFR operating conditions for these experiments (i.e., precursor concentrations, ozone concentrations), is there potential for the interfering α-pinene + ozone reaction to contribute significantly to SOA yields?

Response: According to our previous study (Liggio et al., 2016), α-pinene is likely the main SOA precursor for background OA in the OS region. We have added the following content in the revised manuscript for clarity (P3, L18-20): "Alkanes are the main component of OS emissions, while α-pinene is a representative biogenic precursor which likely contributes significantly to the background SOA observed in OS region (Liggio et al., 2016)."

We have also added "Under the operating conditions used here for α-pinene experiments, OH reaction contributes 64%-98% of the α-pinene gaseous loss across the entire OH exposure range, and >90% after 3 equivalent days, with α-pinene + $O_3$ reaction playing a minor role" in P8, L8-10 of the revised manuscript.

[2] Wall losses (Section 3.1.1): The authors state that vapor wall losses are likely minimal based on the diffusion timescale relative to the residence time within the reactor, then state the critical assumption that flow in the reactor is ideally laminar. Is this assumption solely based on fluid dynamics information from previously designed OFRs? The authors cite CFD done by Huang et al. (2017) for the CPOT on page 4 (lines 6-7) to justify the assumption, but I'm curious as to how the differences between the ECCC-OFR and the CPOT would change the fluid dynamics. For example, the ECCC-OFR has a straight outlet rather than a conical one like the CPOT. Is there potential for jetting or dead volume around the outlet? What are the benefits to sampling from the center line?

Response: We have now performed CFD simulations on the ECCC-OFR and included these results in the Supplement (Sect. S2).

To assess the near laminar flow of the ECCC-OFR, computational fluid dynamics (CFD) simulations were performed using ANSYS Fluent software (Version 2019 R2) in three dimensions to characterize the flow field inside the ECCC-OFR. Hybrid tetrahedral–hexahedral mesh consisting of $5.7 \times 10^5$ computation cells were used. Turbulence was modeled using a realizable k-epsilon model. The simulation results are shown in Fig. S4. It is shown in Fig. S4a that the flow velocity distribution in the reactor is generally uniform. A high velocity is observed only near the inlet, but reduces to the average velocity in the conical diffuser. The velocity distribution here indicates that jetting is much weaker in ECCC-OFR compared to PAM (Mitroo et al., 2018). Fig. S4b indicates that the flow field is quite good in ECCC-OFR, with a small recirculation zone, similar to previous studies using a conical diffusion inlet (Huang et al., 2017;Ihalainen et al., 2019), but much better than PAM (Mitroo et al., 2018).

[Figure]

Figure S4. CFD simulation results: (a) velocity distribution; (b) vectors showing flow field. The red lines in (b) indicate the areas with recirculation.

Based upon the CFD simulation results above, we know that the flow field in ECCC-OFR is not perfectly ideal laminar flow, though it is significantly better than previous OFRs with a straight inlet, e.g., PAM (Mitroo et al., 2018). Hence, our assumption based on ideal laminar flow (using a diffusion timescale

compared to the residence time to infer the gas-wall interactions) was removed in our revised paper (P7, L16-26).

From the CFD simulation results above, we also know that there is no jetting or dead volume around the sampling outlet. The non-laminar flow at the end of the OFR only influences the side flow, not the sampling flow.

The benefits to sampling from the centerline is the minimization of the interactions with walls (Lambe et al., 2011), as most of the flow that interacts with the walls exit from the side outlets.

**Technical Comments**

[1] Page 2, Lines 19-20: The authors state that a single species approach to studying SOA formation is "impractical." To me, "impractical" implies some sort of logistical difficulty and sells the point short. I'd consider reframing this sentence to emphasize atmospheric relevance for the OS, which is critical to consider when performing lab studies.

Response: We changed the sentence into "As a result, using a single species approach to studying SOA formation from OS is unrepresentative." (P2, L19-20).

[2] Page 2, Lines 21-22: Consider restructuring this sentence for clarity. Perhaps "Precursor emissions occur throughout the OS surface mining and processing production cycle, and they originate from sources including…"

Response: This sentence was modified to be "Precursor emissions occur throughout the OS surface mining and processing production cycle, and they originate from sources including open pit surface mines, processing plants and tailings ponds" (P2, L21-22).

[3] Page 2, Line 24: Define "OH" as "hydroxyl radicals (OH)" before using the abbreviation.

Response: Revised (P2, L24).

[4] Page 2, Line 28: "Complimentary" should be "complementary." This spelling should also be changed on page 3, line 13.

Response: Revised (P2, L28; P3, L13).

[5] Page 3, Line 6: Replace the semicolon after "vary" with a comma.

Response: Revised (P3, L6).

[6] Page 4, Line 11: Replace "Hg" with "mercury."

Response: Revised (P4, L12).

[7] Page 5, Line 5: Define the THC acronym here.

Response: Revised (P5, L17).

[8] Figure 1: Consider matching the color of the top and right axes to the alkane data points to visually distinguish the gas-phase data from the particle-phase data.

Response: The Figure 1 (now is Figure 2) is revised to be:

[Figure]

Figure 2. Particle (left and bottom axis) and gas (right and top axis) transmission efficiencies ($P_{trans}$ and $G_{trans}$) for the ECCC-OFR. Particle transmission efficiencies of other OFRs are shown for comparison: PAM-glass and TPOT (Lambe et al., 2011), PAM-metal (Karjalainen et al., 2016), TSAR (Simonen et al., 2017), CPOT (Huang et al., 2017) and PEAR (Ihalainen et al., 2019).

[9] Page 8, line 9: The sentence starting with "This despite" is not a full sentence.

Response: This sentence was merged with previous sentence to be "…for unseeded experiments, despite initial concentrations of…" (P9, L5).

[10] Page 10, line 21: Replace the semicolon after "mixtures" with a comma.

Response: Revised (P11, L9).

[11] Page 10, line 25: It would be helpful to cite the specific section in supporting information so the reader can easily flip to it as needed.

Response: It was modified to be "… as described in detail in Sect. S5 of the Supplement" (P11. L13).

[12] Figure 4a and 4b: Consider emphasizing the different y axis scales between the two panels in either the text or the figure caption. Otherwise, the differences between seeded and non-seeded results can be difficult to pick out visually.

Response: "Note that the y-axis ranges are different in (a), (c), and (d)" was added in the figure caption (P24, L8).

[13] I would be interested to see the AMS mass spectra for each OS-related oxidation experiment, perhaps in the supplement.

Response: The AMS mass spectra for each OS-related oxidation experiment are shown in Figure S9. We have added "Although these precursors have very different SOA yields, their AMS mass spectra (Fig. S9) are similar, indicating a similar main precursor composition (alkanes)" at P11, L25-26 of the revised manuscript.

[Figure]

Figure S9. Representative AMS mass spectra for each OS-related oxidation experiment at OH exposure of ~1.5 × $10^{12}$ molec cm$^{-3}$ s.

**References**

Huang, Y., Coggon, M. M., Zhao, R., Lignell, H., Bauer, M. U., Flagan, R. C., and Seinfeld, J. H.: The Caltech Photooxidation Flow Tube reactor: design, fluid dynamics and characterization, Atmospheric Measurement Techniques, 10, 839-867, 10.5194/amt-10-839-2017, 2017.

Ihalainen, M., Tiitta, P., Czech, H., Yli-Pirilä, P., Hartikainen, A., Kortelainen, M., Tissari, J., Stengel, B., Sklorz, M., Suhonen, H., Lamberg, H., Leskinen, A., Kiendler-Scharr, A., Harndorf, H., Zimmermann, R., Jokiniemi, J., and Sippula, O.: A novel high-volume Photochemical Emission

Aging flow tube Reactor (PEAR), Aerosol Science and Technology, 53, 276-294, 10.1080/02786826.2018.1559918, 2019.

Karjalainen, P., Timonen, H., Saukko, E., Kuuluvainen, H., Saarikoski, S., Aakko-Saksa, P., Murtonen, T., Bloss, M., Dal Maso, M., Simonen, P., Ahlberg, E., Svenningsson, B., Brune, W. H., Hillamo, R., Keskinen, J., and Rönkkö, T.: Time-resolved characterization of primary particle emissions and secondary particle formation from a modern gasoline passenger car, Atmospheric Chemistry and Physics, 16, 8559-8570, 10.5194/acp-16-8559-2016, 2016.

Lambe, A. T., Ahern, A. T., Williams, L. R., Slowik, J. G., Wong, J. P. S., Abbatt, J. P. D., Brune, W. H., Ng, N. L., Wright, J. P., Croasdale, D. R., Worsnop, D. R., Davidovits, P., and Onasch, T. B.: Characterization of aerosol photooxidation flow reactors: heterogeneous oxidation, secondary organic aerosol formation and cloud condensation nuclei activity measurements, Atmospheric Measurement Techniques, 4, 445-461, 10.5194/amt-4-445-2011, 2011.

Liggio, J., Li, S. M., Hayden, K., Taha, Y. M., Stroud, C., Darlington, A., Drollette, B. D., Gordon, M., Lee, P., Liu, P., Leithead, A., Moussa, S. G., Wang, D., O'Brien, J., Mittermeier, R. L., Brook, J. R., Lu, G., Staebler, R. M., Han, Y., Tokarek, T. W., Osthoff, H. D., Makar, P. A., Zhang, J., Plata, D. L., and Gentner, D. R.: Oil sands operations as a large source of secondary organic aerosols, Nature, 534, 91-94, 10.1038/nature17646, 2016.

Mitroo, D., Sun, Y., Combest, D. P., Kumar, P., and Williams, B. J.: Assessing the degree of plug flow in oxidation flow reactors (OFRs): a study on a potential aerosol mass (PAM) reactor, Atmospheric Measurement Techniques, 11, 1741-1756, 10.5194/amt-11-1741-2018, 2018.

Simonen, P., Saukko, E., Karjalainen, P., Timonen, H., Bloss, M., Aakko-Saksa, P., Rönkkö, T., Keskinen, J., and Dal Maso, M.: A new oxidation flow reactor for measuring secondary aerosol formation of rapidly changing emission sources, Atmospheric Measurement Techniques, 10, 1519-1537, 10.5194/amt-10-1519-2017, 2017.